



# Drivers of soil organic carbon from temperate to alpine forests: a model-based analysis of the Swiss forest soil inventory with Yasso20

Claudia Guidi[1*], Sia Gosheva-Oney[1,2*], Markus Didion[1], Roman Flury[1], Lorenz Walthert[1], Stephan Zimmermann[1], Brian J. Oney[1], Pascal A. Niklaus[2], Esther Thürig[1], Toni Viskari[3,4], Jari Liski[3], Frank Hagedorn[1]

[1]Swiss Federal Institute for Forest, Snow and Landscape Research WSL, Birmensdorf, Switzerland

[2]Department of Evolutional Biology and Environmental Studies, University of Zurich, Switzerland

[3]Finnish Meteorological Institute, Helsinki, Finland

[4]Joint Research Centre, Ispra, Italy

*These authors contributed equally to this study

Correspondence to: Claudia Guidi (claudia.guidi@wsl.ch)

**Abstract.** Predicting soil organic carbon (SOC) stocks and its dynamics in forest ecosystems is crucial for assessing forest C balance, but the relative importance of key controls - litter inputs, climate, and soil properties - remains uncertain. Here, we linked SOC stocks at 556 old-growth Swiss forest sites from 350 to 2000 m a.s.l. to a comprehensive set of environmental variables, encompassing climate (mean annual precipitation, MAP: 700-2100 mm, mean annual temperature, MAT: 0-12°C), soil properties, and forest types. In addition, we compared measured SOC stocks with stocks simulated by the Yasso20-model that is widely used for reporting SOC stock changes. Since Yasso20 is driven solely by litter inputs and climate, deviations between modelled and measured stocks can reveal the significance of additional factors such as organo-mineral interactions that we hypothesized to be crucial for SOC stocks.

Total SOC stocks exhibited distinct regional patterns, with the highest values in the Southern Alps, where soils are rich in Fe and Al oxides and receive high MAP. On average, total SOC stocks simulated by Yasso20 aligned well with measured SOC stocks (13.7 vs 13.2 kg C m$^{-2}$). However, the model did not capture regional SOC variability, underestimating SOC stocks by up to 7 kg C m$^{-2}$ in the Southern Alps. The underestimation was primarily explained by soil mineral properties with their influence depending on soil pH. In soils with pH ≤ 5, exchangeable Fe had the strongest effect on Yasso20 deviations from measured stocks, while in soils with pH > 5, exchangeable Ca had the strongest effect on model deviations. Beyond Fe and Ca, MAP emerged as an important driver of total SOC stocks, with SOC stocks increasing with MAP. At higher elevation, this coincided with low MAT and a high share of conifers. While Yasso20 accounted for MAT, Yasso20 underestimated SOC stocks for MAP > 1400 mm.



Overall, our results indicate that mineral-driven SOC stabilization and climate are the primary drivers of Yasso20 deviations

from measured SOC stocks. Incorporating mineral-driven SOM stabilization and coupling to a soil water model can improve the modeling of SOC stocks. However, further studies are needed to verify how C stabilization mechanisms and soil moisture can be included in model-based estimates of SOC stock changes, which is the primary application of Yasso in greenhouse gas inventories.

## 1 Introduction

Soils are the largest C pool in terrestrial ecosystems, with forest soils storing over 40% of terrestrial ecosystem organic C as soil organic matter (SOM) (Prescott and Grayston, 2023). The complex nature of SOM – comprising a heterogenous mixture of components that turn over on daily to millennial time scales (Sierra et al., 2017; Van Der Voort et al., 2017) – and of processes driving SOM stabilization (Schmidt et al., 2011; Kleber et al., 2015; Lehmann and Kleber, 2015) make it difficult to accurately estimate and predict soil organic carbon (SOC) stocks and its responses to environmental changes

(Schrumpf et al., 2011; Smith et al., 2020).

SOC storage is considered to depend mostly on the C input to soils and its transformation and stabilization processes, which are controlled by environmental and biological drivers (Chen et al., 2013; Angst et al., 2018). In temperate forests, SOC storage was shown to be linked to climate, with greater SOC stocks in cool, humid mountainous regions and smaller stocks in warmer and drier regions, as shown in Bavaria (Wiesmeier et al., 2013) and Switzerland (Gosheva et al., 2017). In

the German Alps, the increased temperatures during the last three decades have driven topsoil organic C losses in forests with larger losses at low-elevation sites (Prietzel et al., 2016). However, geochemical factors may exert a strong additional control on SOC stocks by binding SOC to mineral surfaces, leading to long-term stabilization and SOC accumulation (Hagedorn et al., 2003; Doetterl et al., 2015; Doetterl et al., 2018; Reichenbach et al., 2023). In temperate forest soils, the type and reactivity of soil minerals interacting with SOM were shown to be the principal factor driving SOC accumulation

and formation of mineral-associated organic matter, with oxide-dominated soils having a higher capacity to accumulate soil C than soils dominated by phyllosilicate clays (Bramble et al., 2023). For carbonate-containing soils, SOC interactions with $Ca^{2+}$, occlusion within aggregates in Ca-rich substrates and inclusion into carbonates, have been identified as primary SOC stabilization mechanisms (Rowley et al., 2018). In addition to geochemical factors, SOC accumulation is modulated by forest productivity, management intensity and tree species (Bramble et al., 2023), which affect the quantity and quality of

litter inputs entering the soil (Vesterdal et al., 2013; Mayer et al., 2020).

While process-based models increasingly integrate SOM-mineral interactions (Abramoff et al., 2018; Abramoff et al., 2022; Brunmayr et al., 2024), SOM stabilization processes are poorly incorporated into models that are used for greenhouse gas (GHG) accounting of the forest soil C balance, which are generally based on data from National Forest Inventories (Didion et al., 2016; Hernández et al., 2017). Reasons include the poor quantitative knowledge of the complex processes

driving SOM storage, and the limited data availability of soil properties at national scale. Simpler soil C models, in contrast,





typically assume that SOC storage is primarily determined by the quantity and quality of litter inputs and climatic conditions, regulating decomposition and stabilization of organic matter (Liski et al., 2005; Ågren et al., 2008). This facilitates their broader applicability. In several European countries including Switzerland, GHG reporting of SOC stock changes in forests is based on model simulations using the soil C cycling model Yasso (Tuomi et al., 2009; Didion et al., 2016; Hernández et al., 2017). The Yasso model was originally developed for forestry applications, relying on data available in forest inventories and basic climate data, without incorporating soil properties (Liski et al., 2005; Tuomi et al., 2009). Currently, the new version of the model (Yasso20) has been calibrated with a more advanced method and using a global SOC dataset (> 4000 measurements), which resulted in an overall better model performance compared to Yasso07 (Viskari et al., 2022).

Here, we aimed to (i) identify the factors controlling SOC stocks in Swiss forest soils, and to (ii) test the ability of Yasso20 to simulate SOC stocks in forest soils spanning a large gradient of climate, soil biogeochemistry and forest types across Switzerland. Our main approach was to make use of the deviations between Yasso-simulated (i.e. driven only by litter inputs and climate) and measured SOC stocks to infer the importance of mineral-driven SOC stabilization. For this, we analyzed a comprehensive dataset including measured SOC stocks in the organic layers and mineral soil up to 100 cm depth, as well as soil physico-chemical properties for 556 soil profiles located in forests older than 120 years (Gosheva et al., 2017), with an elevation ranging from 370 to 1960 m a.s.l. The dataset covered a wide range of climatic conditions (mean annual precipitation of 700-2100 mm, mean annual temperature 0-12 °C), soil properties (i.e. pH 3-8) and forest types. The dataset included soil physico-chemical characteristics such as clay content, pH, exchangeable Fe, Al, and Ca, topographic attributes, tree species composition, and satellite-derived NPP. Given that the soil C model Yasso20 accounts only for litter inputs and climatic conditions, we hypothesised that mineral soil components driving SOM stabilization (e.g. Fe, Al, and Ca content) would largely explain the deviations between simulated and measured SOC stocks. In turn, this would imply that mineral-driven C stabilization is a key process for soil C storage at the regional and national scale which should be considered in models used for GHG reporting.

## 2 Methods

### 2.1 Study area and sampling sites

For our study, we considered 556 sites in old-growth forests i.e. forests older than 120 years (Fig. 1; 52% broadleaf- vs 48% conifer-dominated sites) from a soil database encompassing approximately 1000 soil profiles (Gosheva et al., 2017). In combination with the typically small-scale, single tree-based management in Switzerland, the focus on old forests ensures that there have been only minimal disturbances in the forest cover over the past decades. Forests in Switzerland can be divided into five major biogeographic regions (Fig. 1) with specific soil properties and forest types (Gosheva et al., 2017).

Most of our sites are located in the Swiss Plateau ($n = 164$), a region characterized by more or less deeply decarbonated soils from still calcareous moraines and tertiary sediments (molasse), followed by sites in the Pre-Alps ($n = 138$), a region consisting of soils with highly variable weathering and decalcification depths from very different types of sediments (Gnägi





and Labhart, 2015). The remaining sites are in the Alps ($n = 81$) on heterogeneous bedrock, in the Jura ($n = 54$) dominated by limestone or marl, and in the Southern Alps ($n = 31$) with mainly gneiss as bedrock (Gnägi and Labhart, 2015).

Waterlogged soils (i.e. redoximorphic soils, characterized by the periodic or permanent oxygen shortage) were studied separately ($n = 88$), since excess moisture dominates soil development and SOM stabilization mechanisms at these soils.

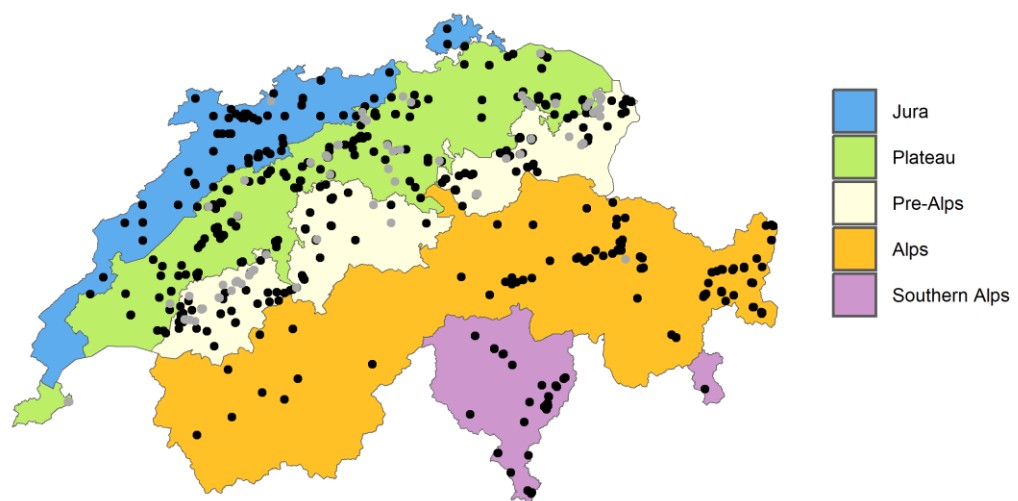

**Figure 1: Distribution of the sampled forest sites (total *n* sites = 556, of which *n* = 88 waterlogged soils shown in grey) in the five biogeographic regions of Switzerland. Note that the Alps have fewer sites since largely un-forested (i.e. above the treeline).**

**2.2 Soil sampling and analysis, climate and topography**

The 556 soil profiles were sampled from 1989 to 2004 by genetic horizons up to the parent material. Composite soil samples were taken from the front wall of the soil profiles (over an average width of 70 cm). Samples were dried at 60°C and sieved with a 2-mm mesh before chemical analysis. Soil type was classified according to the World Reference Base (Iuss, 2007).

Total and organic C contents were measured in ground samples by dry combustion using an elemental analyzer (NC 2500, CE Instruments, Italy). Soil samples with pH > 6 were first treated with HCl to remove inorganic C prior to dry combustion (Walthert et al., 2010). Soil texture was characterized by measuring the clay, silt and sand contents using the sedimentation method according to Gee and Bauder (1986). Soil pH was measured potentiometrically in a 0.01 M CaCl$_2$ solution. Contents of exchangeable Al, Fe, and Ca (in mmolc kg$^{-1}$) were obtained by 1 M ammonium chloride extraction

(Walthert et al., 2004). Soil properties were calculated for 0-30 cm and 0-100 cm depth intervals of the mineral soil by weighted averages of their contents according to the amount of fine earth in the various soil horizons.

Data on elevation and slope were determined from a 25-m digital elevation model for all sites (Swisstopo, 2011). Slope orientation and topography were assessed during the soil surveys. Annual climate data for each site (mean monthly



temperature and mean monthly precipitation for the period 1961-1990) were obtained from the gridded data produced by the
Swiss Federal Office of Meteorology and Climatology (Meteoswiss, 2024). The Braun-Blanquet cover abundance scale
(Braun-Blanquet, 1964) was used to quantify plant species cover in an area ranging from 100 to 500 m$^2$ (Walthert et al.,
2013). Percentage of broadleaf tree species was calculated as the sum of the cover of all broadleaf species divided by the
sum of the cover of all tree species at the tree level. The forest sites were subdivided into two types based on the broadleaf
percentage: coniferous (0-50%) and broadleaf (51-100%) forests.

## 2.3 Calculation of SOC stocks from sampled soil profiles

SOC stocks were calculated separately for the organic layers including L (undecomposed litter), F (fermentation) and H
(humified) horizons and the mineral soil at 0-100 cm depth. Total SOC stock is the sum of stocks in the organic layers and in
the mineral soil. SOC stocks in the organic layer were calculated according to Hagedorn et al. (2010), with the mass of the
organic layer calculated as the product of the density (L: 0.10 g cm$^{-3}$, F: 0.15 g cm$^{-3}$, H: 0.20 g cm$^{-3}$) and the volume (based
on measured thickness), multiplied by the percentage of C content. Mineral SOC stocks were calculated using the following
Eq. (1):

$$SOC(h, z) = \sum_1^z (h_i (1 - \theta_i) \rho_i C_i) \qquad (1)$$

where $SOC(h,z)$ represents SOC stocks (kg C m$^{-2}$) of all the $z$ mineral soil horizons, $C_i$ - the organic carbon content of the
horizon $i$ (kg kg$^{-1}$ of C), $\rho_i$ is the density of the fine earth (g cm$^{-3}$), $\theta_i$ is the volumetric stone content (m$^3$ m$^{-3}$), and $h_i$ is the
horizon thickness (m). A pedotransfer function (PTF) based on a calibration dataset of 559 mineral soil horizons from 134
different Swiss forest sites and a validation set of 131 horizons from 34 sites, was used to estimate the density of the fine
earth fraction (Nussbaum et al., 2016). Covariates used in this PTF are sampling depth, slope, field estimates of stone content
and soil density, biogeographic region, and organic C content.

## 2.4 Input derivation and SOC stocks simulations with Yasso20

SOC stock simulations were conducted with the soil C cycling model Yasso20. The model was calibrated by Viskari et
al. (2022) using several datasets describing different processes of the soil C cycle, including litterbag decomposition time
series, a woody decomposition dataset and global SOC stock measurements, which make Yasso20 potentially suitable for a
wide range of environmental conditions. Based on litter input data (i.e., foliage, deadwood etc.), Yasso20 simulates flows of
C between four chemical classes of C compounds and a humus ("H") pool representing long-lived, stable SOC as a function
of temperature and precipitation. Litter inputs are sub-divided into the following four different chemical C compounds that
are the same as the SOC classes in the model: "A" represents hydrolysable compounds in acid, "W" water-soluble
compounds, "E" ethanol-soluble compounds, and "N" non-soluble compounds. Each of these compounds have different C
decomposition rates. The sum of the AWEN compound classes (representing the "labile" C pools) and the H pool (the
"stable" C pools) corresponds to the total "simulated" SOC stocks, including the SOC stored in the organic layers plus in the





mineral soil up to 100 cm depth. Parameters defining C decomposition rates and C cycling between C compartments and to a more stable H pool are obtained probabilistically using Markov Chain Monte Carlo sampling (Viskari et al., 2022) and are fitted based on data from several litterbag studies (Berg et al., 1991; Trofymow, 1998; Gholz et al., 2000), woody litter decomposition experiments (Mäkinen et al., 2006), and global SOC measurements from Oak Ridge National Laboratory (Zinke et al., 1986).

To obtain a proxy of site-specific litter inputs for Yasso20 simulations, we derived the average net primary production (NPP) for the period 2001-2022 from Terra and Aqua MODIS-satellite at 500-m resolution (Running and Zhao, 2021a, b), with a maximum NPP to GPP (gross primary production) ratio of 0.5 (Viskari et al., 2022). We partitioned the NPP into broadleaf and conifer species, multiplying the NPP by the percentage of broadleaf and conifer species recorded by field assessments at each site. Then, we divided the NPP into different tree components, multiplying the NPP by average tree

allocation factors (stem: 0.30; branches: 0.04; twigs: 0.03; coarse roots: 0.11; fine roots: 0.15; foliage: 0.27; seeds: 0.10) derived from 15 years of tree growth data at 18 long-term forest ecosystem research (LWF) sites across Switzerland (Etzold et al., 2014). Fine roots and foliage represented the group of non-woody litter, whereas woody litter included stem wood, branches, twigs, coarse roots, and seeds. The C inputs for each pool were separated into the four AWEN compounds following experimentally-derived fractions at LWF sites in Switzerland (Didion et al., 2014). Data for observed climate were

obtained for each site (30-year average climate, 1961-1990) from spatially gridded data of the Federal Office of Meteorology and Climatology MeteoSwiss (Meteoswiss, 2024), see *section 2.2*.

    The Yasso20 simulations were performed in R ([www.r-project.org](www.r-project.org)) version 4.2.2 (R Core Team, 2022) with the package *Ryassofortran*, version 0.4.0 (Pusa, 2023). Average litter input derived from NPP and climate data were used with spin-up simulations to reach theoretical steady-state SOC stocks (Mao et al., 2019), assuming that current SOC stocks have

accumulated over centuries (Gimmi et al., 2013). At each site, independent simulations were based on 500 randomly-sampled parameter vectors to represent uncertainty related to model parameters (Viskari et al., 2022).

## 2.5 Statistical analysis

    All statistical analyses were performed with R version 4.2.2 (R Core Team, 2022). To summarize the correlated numerical explanatory variables into larger groups, we performed a principal component analysis (PCA) including all the

numerical variables (pH, clay content, exchangeable contents of Fe, Al, and Ca, mean annual temperature or MAT, mean annual precipitation or MAP, NPP, percentage of broadleaves and slope) with scaled variables. Only principal components with eigenvalues >1 were retained (Kaiser-Guttman criterion).

    The effects of (i) main principal components, and (ii) soil physico-chemical properties (pH, clay content, exchangeable contents of Fe, Al, and Ca) in the upper 30 cm mineral soil - the most relevant depth for tree rooting, litter decomposition

processes and organic layer development - and site variables (MAT, MAP, NPP, percentage of broadleaves and slope) on total SOC stocks and Yasso20 deviations (simulated minus measured values of total SOC stocks) were tested using linear mixed-effect models with the *nlme* package, version 3.1–160 (Pinheiro et al., 2022). In order to account for a small spatial



autocorrelation in the model residuals, the biogeographic region was included as a random intercept in the model, which allowed to estimate an overall effect of the linear model (i.e. model estimate). Effects of explanatory variables on SOC
stocks were also tested for each biogeographic regions by linear models. To meet the assumption of normally distributed residuals, the numerical explanatory variables were log- or square-root transformed when necessary, which ensured a linear relationship between the explanatory and the response variable. Then, all explanatory variables were standardized to a mean of 0. The assumption of normal distribution of residuals was verified through quantile–quantile plots and plots of residuals vs fitted values. Based on existing literature, the variables pH, clay content, exchangeable Fe, Al, and Ca, MAT, MAP, NPP,
percentage of broadleaves and slope were identified as key drivers of SOC stocks and thus as explanatory variables. Exchangeable Al was excluded from the final statistical model due to its strong correlation with pH and exchangeable Fe ($r$ = -0.93 and +0.87, respectively). Exchangeable Fe was retained as the primary proxy for pedogenic oxides (Fig. S4). For the statistical analysis of Yasso20 deviations, NPP was excluded from the final model since it was used as main input of the soil C cycling model Yasso20. Two-way interactions between climate and soil properties (i.e. exchangeable contents of Fe, Ca,
clay content, and pH) and between soil properties and pH were also tested (Table S2, Table S3), since interactions of climate and geochemical factors are known to drive SOC storage (Doetterl et al., 2015).

Measured vs simulated SOC stocks by biogeographic regions and forest types were compared using Welch's *t*-tests, after verifying normality assumptions. Correlations between simulated and measured SOC stocks were tested using Pearson's product moment correlation coefficient (r), which agreed well with Spearman's rank correlation coefficient (*not shown*).

## 195 3 Results

### 3.1 Site characteristics

Distinct site characteristics were found in each of the five biogeographic regions of Switzerland (Fig. 1, Table 1). Topsoils in the Jura showed the highest pH, as well as contents of clay and Ca, while the most acidic soils were found in the Southern Alps, with highest contents of exchangeable Fe, oxalate-extractable Fe and Al (Table 1). MAP was highest in the
Southern Alps and the Pre-Alps, whereas the Plateau had the highest MAT (Table 1). Under conifer-dominated forests, topsoils were more acidic (average pH ± standard error at 0-30 cm depth, conifers: 4.7 ± 0.1 vs broadleaves 5.2 ± 0.1) and had higher contents of exchangeable Fe and Al than broadleaf-dominated forests (*data not shown*).





**Table 1: Average values of soil properties (pH, clay content, exchangeable contents of Fe, Al, and Ca) in the upper 30 cm of mineral soil, annual climate data (MAT = mean annual temperature, MAP = mean annual precipitation), net primary production (NPP) from MODIS (average 2001-2022), and elevation in the 556 sites across the five biogeographic regions of Switzerland, with waterlogged soils shown separately. For a subset of sites ($n$ = 123), oxalate-extractable Fe and Al (Fe$_{ox}$ and Al$_{ox}$) in the upper 30 cm of mineral soil are reported. Values are means with standard errors in brackets.**

|  | Jura | Plateau | Pre-Alps | Alps | Southern Alps | Waterlogged | Switzerland |
|---|---|---|---|---|---|---|---|
| *All sites* | *n = 54* | *n = 164* | *n = 138* | *n = 81* | *n = 31* | *n = 88* | *n = 556* |
| pH | 6.2 (0.2) | 4.7 (0.1) | 4.8 (0.1) | 5.3 (0.2) | 4.3 (0.2) | 5.3 (0.2) | 5.0 (0.1) |
| Clay (%) | 35.5 (1.9) | 19.6 (0.7) | 24.0 (1.0) | 17.2 (1.1) | 12.8 (1.7) | 25.3 (1.2) | 22.4 (0.5) |
| Fe$_{exch}$ (mmolc kg$^{-1}$) | 0.2 (0.1) | 0.5 (0.1) | 0.9 (0.1) | 0.6 (0.1) | 1.3 (0.3) | 0.6 (0.1) | 0.7 (0.0) |
| Al$_{exch}$ (mmolc kg$^{-1}$) | 7.2 (2.2) | 28.7 (2.3) | 39.6 (4.3) | 18.0 (2.8) | 35.1 (4.9) | 21.8 (4.1) | 27.0 (1.6) |
| Ca$_{exch}$ (mmolc kg$^{-1}$) | 302.0 (23.6) | 73.1 (8.3) | 116.6 (12.3) | 122.5 (11.8) | 48.9 (17.9) | 138.2 (12.1) | 122.3 (6.0) |
| MAT (°C) | 7.0 (0.2) | 8.0 (0.0) | 6.4 (0.1) | 3.9 (0.3) | 6.6 (0.5) | 7.0 (0.2) | 6.7 (0.1) |
| MAP (mm) | 1253 (30) | 1141 (11) | 1508 (22) | 1021 (28) | 1579 (47) | 1329 (28) | 1279 (12) |
| NPP (kg C m$^{-2}$ yr$^{-1}$) | 0.64 (0.01) | 0.65 (0.01) | 0.61 (0.01) | 0.45 (0.01) | 0.57 (0.03) | 0.63 (0.01) | 0.60 (0.01) |
| Elevation (m) | 789 (31) | 592 (10) | 906 (24) | 1427 (41) | 1027 (87) | 789 (34) | 866 (16) |
|  |  |  |  |  |  |  |  |
| *Subset sites* | *n = 10* | *n = 52* | *n = 21* | *n = 12* | *n = 11* | *n = 17* | *n = 123* |
| Fe$_{ox}$ (g kg$^{-1}$) | 2.23 (0.32) | 2.79 (0.22) | 4.38 (0.77) | 5.89 (1.28) | 9.13 (2.68) | 4.43 (0.46) | 4.11 (0.36) |
| Al$_{ox}$ (g kg$^{-1}$) | 2.18 (0.36) | 2.05 (0.23) | 2.68 (0.40) | 2.23 (0.47) | 6.52 (1.47) | 1.99 (0.23) | 2.58 (0.22) |

NPP peaked at an elevation of about 700 m a.s.l., then decreased with increasing elevation (linear model fit for elevation > 700 m: -0.26±0.01 kg C m$^{-2}$ yr$^{-1}$ per 1000 m elevation gain; $R^2$ = 0.65, $P$ < 0.001, Fig. 2a). In contrast to NPP, SOC stocks in the organic layer increased with elevation (3.5±0.4 kg C m$^{-2}$ per 1000 m increase in elevation; $R^2$ = 0.14, $P$ < 0.001; Fig. 2b) in sites excluding waterlogged soils ($n$ = 468). In the mineral soil, SOC stocks were not related to elevation (Fig. 2c), thus the ratio between SOC stored in the organic layer and in the mineral soil increased with elevation. NPP was negatively correlated to organic layer SOC stocks ($R^2$ = 0.08, with Pearson correlation coefficient r = -0.29, $P$ < 0.001), while positively but weakly related to mineral SOC stocks ($R^2$ = 0.01, with Pearson correlation coefficient r = +0.12, $P$ = 0.006).

The first three principal components of the PCA with soil physico-chemical variables and site properties explained 74% of variance in the data (Table S1a). The first PC (PC1) was characterized by high loadings of soil chemical parameters: pH (0.48), Ca (0.47), Al (-0.43), and Fe (-0.40), explaining 35% of the variance. PC2 (24% of variance) showed high loadings of MAT (0.58), NPP (0.58), and broadleaf percentage (0.44), whereas PC3 (15% of variance) of MAP (0.63) and clay content (0.44) (Table S1a).





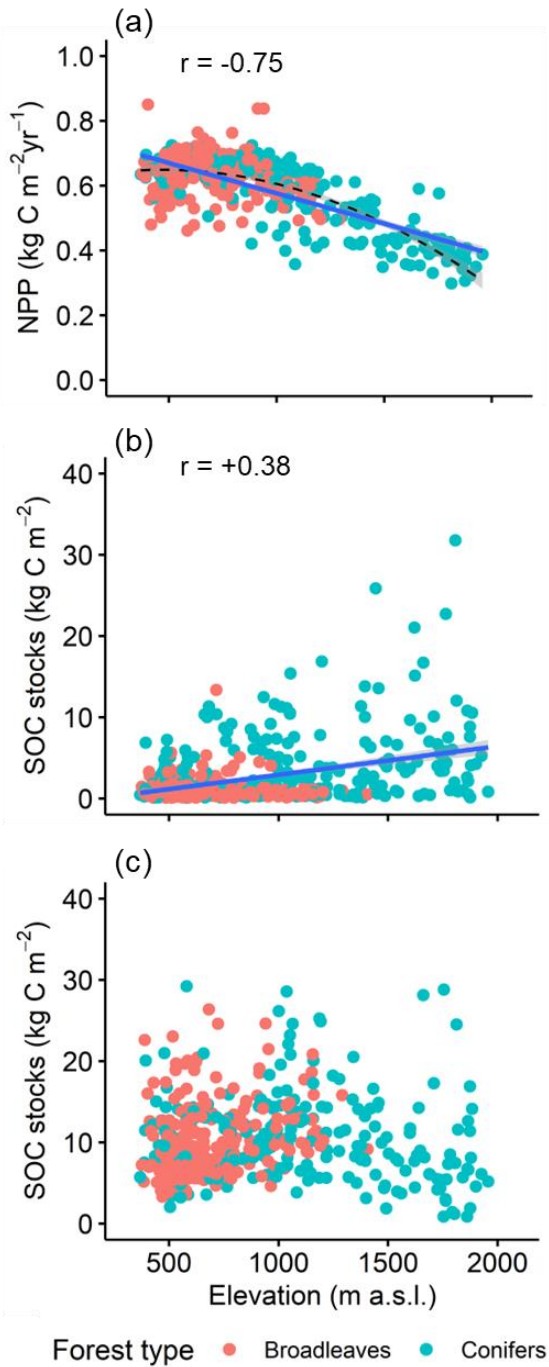

**Figure 2: Net primary production (NPP) from MODIS (average 2001-2022) (a), SOC stocks in the organic layers (b), and in the mineral soil at 0-100 cm depth (c) against elevation in 468 forest sites across Switzerland (excluding waterlogged soils). Plotted lines show significant linear correlations (*P* < 0.05) with 95% confidence intervals in grey and the Pearson correlation coefficient (r). A polynomial regression is also shown as dotted line in (a).**


 

## 3.2 Measured and simulated SOC stocks

Measured SOC stocks were highest in the Southern Alps (19.5±1.7 kg C m⁻²) followed by the Pre-Alps, Jura, and the
Alps, while lowest in the Plateau (10.2±0.3 kg C m⁻², Fig. 3a). On average, the Yasso20-simulated SOC stocks were similar
to the measured stocks at the 556 sites including waterlogged soils (average ± standard error; simulated 13.7±0.1 vs
measured 13.2±0.3 kg C m⁻², Welch's *t*-test: *P* = 0.16). Waterlogged soils had SOC stocks of 15.9±1.2 kg C m⁻², and are thus
considered separately in further analyses. In comparison to measured stocks, simulated SOC stocks differed only little
among the biogeographic regions (Fig. 3a). The largest deviations between simulated and measured SOC stocks were
observed in the Southern Alps, where Yasso20 underestimated stocks by almost 7 kg C m⁻² (-35% of measured SOC stocks,
Welch's *t*-test: *P* < 0.001, Fig. 3a). In contrast, Yasso20 overestimated SOC stocks by about 3 kg C m⁻² in the Plateau (+33%
of measured SOC stocks, *P* < 0.001). The simulations for the Jura, Pre-Alps and Alps agreed well with SOC measurements,
with average deviations below 1 kg C m⁻² (2-4% of measured SOC stocks, *P* > 0.05). In waterlogged soils, Yasso
underestimated SOC stocks by about 2 kg C m⁻² (-13% of measured SOC stocks, *P* = 0.08).

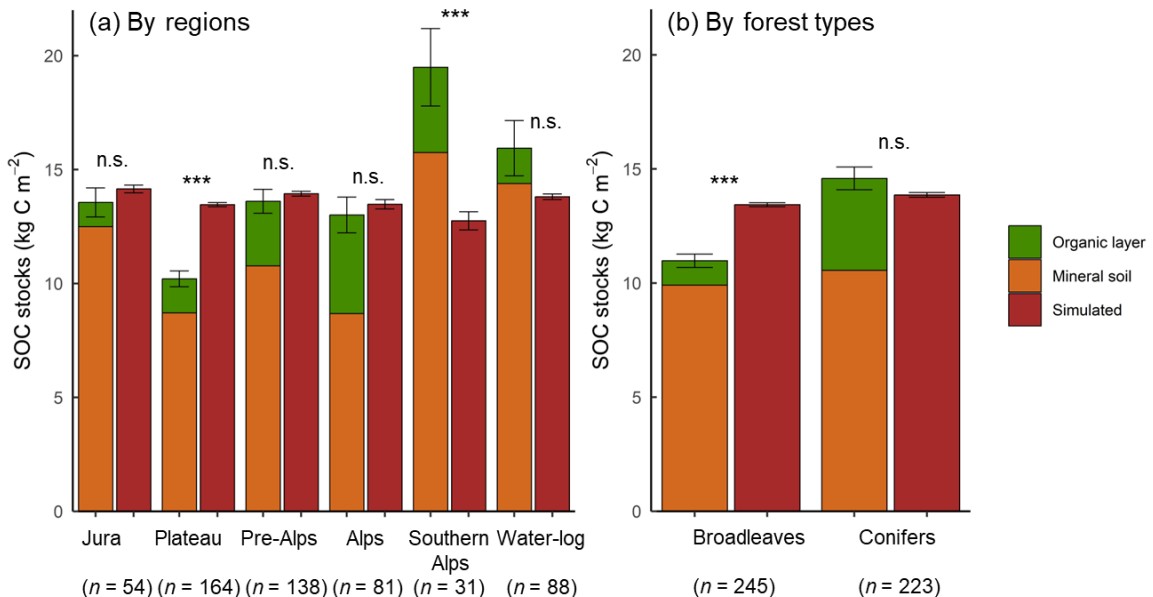

**Figure 3: Comparison between measured SOC stocks vs Yasso20-simulated stocks by: (a) biogeographic regions of Switzerland
(waterlogged soils shown separately), and (b) forest types, excluding waterlogged soils. The measured stocks are shown as organic
layers and mineral soils to 100 cm depth (total *n* sites = 556; excluding waterlogged soils, *n* sites = 468). The simulated stocks at
each site are based on the mean of 500 replicate simulations representing model parameters uncertainty. SOC stocks are
represented as means ± standard errors. *P*-values are calculated with Welch's *t*-tests: *P* ≥ 0.05 (n.s., not significant), *P* < 0.001
(***).**

Measured SOC stocks were greater in conifer- than in broadleaf-dominated stands (14.6±0.5 vs 11.0±0.3 kg C m⁻²
respectively, without waterlogged soils; Fig. 3b), mostly due to higher SOC stocks in the organic layers (+3 kg C m⁻²).
However, Yasso20 estimated similar total SOC stocks in conifer- and broadleaf-dominated forests (13.9±0.1 vs 13.4±0.1 kg





C m$^{-2}$, respectively), overestimating SOC stocks in broadleaf forests by 2.5 kg C m$^{-2}$ (Welch's $t$-test: $P$ <0.001, Fig. 3b) while slightly underestimating SOC stocks in conifer forests by less than 1 kg C m$^{-2}$ ($P = 0.16$, Fig. 3b).

### 3.3 Drivers of SOC stocks and Yasso20 deviations

     SOC stocks were positively correlated with MAP (r = +0.32, $P$ < 0.001, Fig. 4), and negatively correlated with MAT (r

= -0.17, $P$ < 0.001). Since the reactivity of mineral surfaces is pH-dependent, we separated the dataset into soils with pH ≤ 5

and with pH > 5. With pH ≤ 5, SOC stocks increased with increasing content of exchangeable Fe and Al (r = +0.60 and r =

+0.38, respectively, with $P$ < 0.001), while when pH was above 5, SOC stocks increased with increasing exchangeable Ca (r

= +0.49, $P$ < 0.001).

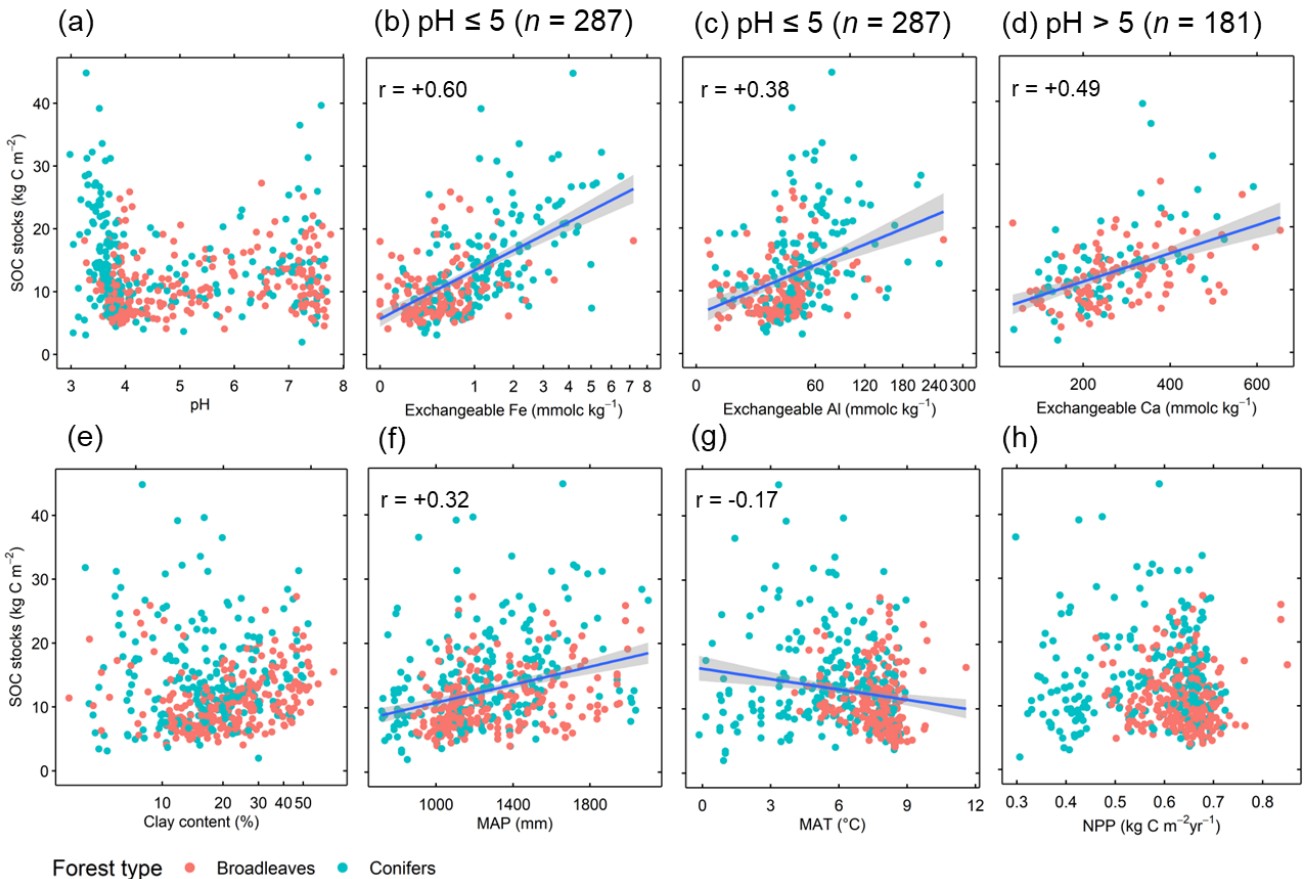

**Figure 4: Correlations between total SOC stocks and selected soil properties (exchangeable Fe and Al are shown on a square-root**
**scale axis, clay on a natural-logarithm scale axis) and site variables (MAP = mean annual precipitation; MAT = mean annual**
**temperature; NPP = net primary production). Total $n$ sites = 468 (waterlogged soils excluded). Plotted lines show significant linear**
**correlations ($P$ < 0.05) with 95% confidence intervals in grey and the Pearson correlation coefficient (r).**



The principal components PC1 (including primarily soil chemical parameters) and PC3 (including MAP and clay) had a

significant effect on SOC stocks (Table S1b). In comparison, PC2 (with MAT, NPP and forest type) was statistically unrelated with SOC stocks (Table S1b). We then tested effects of soil properties and site variables on SOC stocks ($n = 468$ sites, waterlogged soils excluded, Table 2). Linear mixed-effect models, including a random intercept for the region, showed a significant overall effect of exchangeable Fe, pH, MAP, broadleaf percentage and slope on total SOC stocks. We found a positive interaction between pH and exchangeable Ca (Table S2, model with interactions), with exchangeable Ca enhancing

SOC storage at pH > 5 (Fig. S2b). Additionally, there was a significant, negative interaction of clay and MAP (Table S2, Fig. S2a) that mostly reflected SOC-rich soils in the Southern Alps with high MAP but low clay content (Table 1). In soils with pH ≤ 5 ($n = 287$), the exchangeable Fe and MAP had a positive effect on SOC stocks, while slope and clay content had a negative effect on SOC stocks (Table 2). In soils with pH > 5 ($n = 181$), exchangeable Ca and MAP had a positive impact on SOC stocks, while broadleaf percentage and slope decreased SOC stocks. Analysis of drivers of SOC stocks within

biogeographic regions (Table S5) indicated that exchangeable Fe explained the largest proportion of model variance (from 30 to 46% of total $R^2$) for all biogeographic regions except the Jura, characterized by average pH > 6, where exchangeable Ca explained the highest amount of variation (44% of total $R^2$).

**Table 2: Drivers of total SOC stocks. Effects of soil properties in the upper 30 cm mineral soil (i.e. pH, clay content, exchangeable**
**contents of Fe and Ca), MAT (mean annual temperature), MAP (mean annual precipitation), NPP (net primary production), percentage of broadleaves and slope on total SOC stocks (kg C m⁻²). Linear mixed-effect models with region as random intercept were developed for (i) all sites excluding waterlogged soils, (ii) sites with pH ≤ 5, and (iii) sites with pH > 5.**

| | All sites ($n = 468$) | | | | | pH ≤ 5 ($n = 287$) | | | | | pH > 5 ($n = 181$) | | | |
|---|---|---|---|---|---|---|---|---|---|---|---|---|---|---|
| | Est. | SE | t | P | | Est. | SE | t | P | | Est. | SE | t | P |
| (Intercept) | 13.73 | 1.15 | 12.0 | **<0.001** | (Intercept) | 13.09 | 0.84 | 15.6 | **<0.001** | (Intercept) | 14.31 | 1.81 | 7.9 | **<0.001** |
| pH | 1.81 | 0.31 | 5.8 | **<0.001** | pH | 2.03 | 1.23 | 1.6 | 0.10 | pH | -0.24 | 0.55 | -0.4 | 0.67 |
| log(Clay) | 0.12 | 0.59 | 0.2 | 0.84 | log(Clay) | -2.49 | 0.78 | -3.2 | **0.002** | Clay | 0.02 | 0.04 | 0.4 | 0.69 |
| sqrt(Fe) | 6.64 | 0.68 | 9.7 | **<0.001** | sqrt(Fe) | 7.19 | 0.85 | 8.5 | **<0.001** | sqrt(Fe) | 1.53 | 3.43 | 0.4 | 0.66 |
| log(Ca) | 0.20 | 0.25 | 0.8 | 0.42 | log(Ca) | 0.19 | 0.29 | 0.6 | 0.52 | Ca | 0.03 | 0.00 | 5.8 | **<0.001** |
| MAT | -0.02 | 0.21 | -0.1 | 0.92 | MAT | -0.24 | 0.27 | -0.9 | 0.39 | MAT | 0.59 | 0.30 | 2.0 | 0.05 |
| MAP | 0.01 | 0.00 | 5.5 | **<0.001** | MAP | 0.01 | 0.00 | 6.4 | **<0.001** | MAP | 0.00 | 0.00 | 2.6 | **0.010** |
| NPP | 2.09 | 4.37 | 0.5 | 0.63 | NPP | 6.01 | 5.21 | 1.2 | 0.25 | NPP | -13.24 | 6.83 | -1.9 | 0.05 |
| Broadleaf% | -0.02 | 0.01 | -2.9 | **0.004** | Broadleaf% | -0.01 | 0.01 | -1.2 | 0.21 | Broadleaf% | -0.03 | 0.01 | -2.5 | **0.014** |
| sqrt(Slope%) | -0.34 | 0.11 | -3.2 | **0.001** | sqrt(Slope%) | -0.53 | 0.14 | -3.8 | **<0.001** | Slope% | -0.03 | 0.01 | -2.3 | **0.022** |
| DF | 454 | | | | | 273 | | | | | 167 | | | |
| marginal $R^2$ | 0.30 | | | | | 0.48 | | | | | 0.24 | | | |
| conditional $R^2$ | 0.44 | | | | | 0.53 | | | | | 0.57 | | | |
| RMSE (kg C m⁻²) | 4.9 | | | | | 4.6 | | | | | 4.3 | | | |

Model estimates (Est.), standard errors (SE), t statistic and *P*-values are reported (*P* < 0.05 highlighted in bold).
Measurement units of independent variables are reported in Table 1.
DF is the degrees of freedom. Marginal $R^2$ includes the variance of the fixed effects, while conditional $R^2$ both the fixed and random effects calculated with the R package *performance* (Lüdecke et al., 2021).
RMSE is the root mean squared error.



Similarly to SOC stocks, Yasso20 deviations (i.e. difference between Yasso20-simulated and measured SOC stocks, with positive deviations indicating a model overestimation of SOC stocks, while negative deviations a model underestimate, $n = 468$ sites, Table 3) were significantly affected by exchangeable Fe, pH, MAP, broadleaf percentage, and slope. Although the correlation between MAT and the Yasso20 deviations is overall positive ($r = +0.10$, see Fig. S3), the random intercept for the different regions – which accounts for the regional variability - led to a negative estimated linear effect of MAT on

Yasso20 deviations.

**Table 3: Drivers of Yasso20 deviations.** Effects of soil properties in the upper 30 cm mineral soil (i.e. pH, clay content, exchangeable contents of Fe and Ca), MAT (mean annual temperature), MAP (mean annual precipitation), percentage of broadleaves and slope on Yasso20 deviations (simulated minus measured values of total SOC stocks) in kg C m$^{-2}$. Linear mixed-effect models with region as random intercept were developed for (i) all sites excluding waterlogged soils, (ii) sites with pH ≤ 5, and (iii) sites with pH > 5.

| | All sites ($n = 468$) | | | | | pH ≤ 5 ($n = 287$) | | | | | pH > 5 ($n = 181$) | | | |
|---|---|---|---|---|---|---|---|---|---|---|---|---|---|---|
| | Est. | SE | t | *P* | | Est. | SE | t | *P* | | Est. | SE | t | *P* |
| (Intercept) | -0.26 | 1.39 | -0.2 | 0.85 | (Intercept) | 0.42 | 1.02 | 0.4 | 0.68 | (Intercept) | -1.30 | 2.58 | -0.5 | 0.62 |
| pH | -1.78 | 0.32 | -5.6 | **<0.001** | pH | -1.99 | 1.25 | -1.6 | 0.11 | pH | -0.03 | 0.59 | 0.0 | 0.96 |
| log(Clay) | 0.24 | 0.60 | 0.4 | 0.69 | log(Clay) | 2.64 | 0.79 | 3.4 | **<0.001** | Clay | 0.01 | 0.04 | 0.2 | 0.87 |
| sqrt(Fe) | -6.68 | 0.70 | -9.6 | **<0.001** | sqrt(Fe) | -7.24 | 0.86 | -8.4 | **<0.001** | sqrt(Fe) | -4.74 | 3.63 | -1.3 | 0.19 |
| log(Ca) | -0.38 | 0.25 | -1.5 | 0.12 | log(Ca) | -0.34 | 0.29 | -1.2 | 0.24 | Ca | -0.03 | 0.00 | -5.2 | **<0.001** |
| MAT | -0.51 | 0.18 | -2.9 | **0.004** | MAT | -0.43 | 0.24 | -1.8 | 0.07 | MAT | -0.66 | 0.25 | -2.6 | **0.009** |
| MAP | -0.01 | 0.00 | -5.0 | **<0.001** | MAP | -0.01 | 0.00 | -6.2 | **<0.001** | MAP | 0.00 | 0.00 | -1.7 | 0.09 |
| Broadleaf% | 0.03 | 0.01 | 3.2 | **0.0017** | Broadleaf% | 0.02 | 0.01 | 1.5 | 0.14 | Broadleaf% | 0.03 | 0.01 | 2.7 | **0.009** |
| sqrt(Slope) | 0.36 | 0.11 | 3.3 | **0.001** | sqrt(Slope) | 0.52 | 0.14 | 3.6 | **<0.001** | Slope | 0.03 | 0.02 | 2.2 | **0.026** |
| | | | | | | | | | | | | | | |
| DF | 455 | | | | | 274 | | | | | 168 | | | |
| marginal R$^2$ | 0.25 | | | | | 0.43 | | | | | 0.17 | | | |
| conditional R$^2$ | 0.45 | | | | | 0.52 | | | | | 0.66 | | | |
| RMSE (kg C m$^{-2}$) | 5.0 | | | | | 4.7 | | | | | 4.6 | | | |

Model estimates (Est.), standard errors (SE), t statistic and *P*-values are reported (*P* < 0.05 highlighted in bold).
Measurement units of independent variables are reported in Table 1.
DF is the degrees of freedom. Marginal R$^2$ includes the variance of the fixed effects, while conditional R$^2$ both the fixed and random

effects calculated with the R package *performance* (Lüdecke et al., 2021).
RMSE is the root mean squared error.

## 4 Discussion

### 4.1 Soil properties are the primary drivers of SOC stocks

Our study along large environmental gradients from temperate to alpine old forest stands across Switzerland indicated

that soil mineral properties, together with climate, play a dominant role in controlling SOM stabilization and SOC stocks. This finding aligns with a global scale study that demonstrated the primary influence of soil properties, alongside climate, in driving SOC stocks across whole-soil profiles (Luo et al., 2021). Overall, exchangeable Fe was the predictor with strongest effect on SOC stocks within biogeographic regions with acidic topsoil (Table S5). The SOC stocks were greatest in the Southern Alps, having a slightly lower NPP than the Swiss average but higher contents of exchangeable Fe and Al (Table 1).



Moreover, the comparison of SOC stocks simulated with the Yasso20 model - which does not account for physico-chemical soil properties - and measured SOC stocks showed the largest deviations in soils with high content of stabilizing minerals (Fig. 3, Fig. S3). The importance of different cations in stabilizing SOC depended on soil pH – confirmed by the significant interaction between pH and Ca ($P < 0.001$, Table S2, Fig. S2) – which agrees with process-based studies (Rowley et al., 2018). In forest soils with pH $\leq 5$, exchangeable Fe had a significant positive effect on SOC stocks (Table 2) and was the

strongest predictor of SOC stocks in regions with acidic conditions (Table S5), with Yasso20 underestimating SOC stocks at high exchangeable Fe contents (Table 3, Fig. S3). In soils with pH $> 5$, high exchangeable Ca was associated with underestimates of SOC stocks by Yasso20 (Table 3, Fig. S3). Clay content - regarded as a key property for SOM stabilization (Rasmussen et al., 2018) - appeared weakly associated with SOC stocks in Swiss forest soils (Table 2, Figure 4), which is most likely due to the overarching effect of Fe (and Al) driving the high SOC stocks in acidic, sandy soils under

high MAP (Table 1). Although clay normally regulates the SOC storage because of the stabilizing effect of clays on organic compounds (Alvarez and Lavado, 1998; Wiesmeier et al., 2019), we found a positive effect of clay on SOC stocks only in the Plateau region (Table S5). The negative interaction between MAP and clay content ($P < 0.001$, Table S2, Fig. S2a) was likely driven by the high SOC stocks in the Southern Alps, with high MAP, acidic conditions, and sandy soils (Solly et al., 2020).

Due to the low solubility product of Fe-oxides, Fe extracted with $NH_4Cl$ comprises $Fe^{2+}$ and/or organically bound, colloidal Fe(III)-cations (Schwertmann et al., 1987). Exchangeable Fe may, however, serve as an indicator for the content of Fe-oxides as indicated by significant correlations of $NH_4Cl$-extractable Fe with organically-bound Fe (pyrophosphate-extractable, r = +0.78, $P < 0.001$, $n = 62$) and poorly-crystalline Fe-oxides (oxalate-extractable, r = +0.62, $P < 0.001$, $n = 123$) for a subset of surface soils where Fe-oxides have been measured (Fig. S4). Deviations between modeled and measured

SOC stocks were also related to exchangeable Al under acidic conditions (Fig. S3). Since exchangeable Al was strongly correlated to exchangeable Fe (r = +0.87), effect of Al on model deviations could not be disentangled from that of Fe and was thus excluded from the full statistical model. The dominant role of pedogenic oxides for SOM stabilization arises from their large and highly reactive mineral surfaces and positive charge under acidic and neutral conditions (Kaiser and Guggenberger, 2003), which allows interactions with SOM through cation bridging, electrostatic interactions or the

formation of inner- and outer-sphere complexes (Kleber et al., 2015; Rasmussen et al., 2018). Beyond its relation to pedogenic oxides, exchangeable Fe and Al – more often available for large data sets – may represent a proxy for the weathering status of soils (Eimil-Fraga et al., 2015), which is crucial for providing reactive surfaces stabilizing SOM.

In calcareous soils, Ca-mediated SOM stabilization is linked to the ability of $Ca^{2+}$ to bridge negatively charged organic matter surfaces when pH is above neutrality, through inner- and outer-sphere interactions and Ca-mediated aggregation

(Rowley et al., 2018), which limit the decomposer activity and thus lead to preferential stabilization of organic compounds (Gocke et al., 2011; Rowley et al., 2021). However, exchangeable Ca is (similarly to exchangeable Fe for pedogenic oxides) only an indirect measure for $CaCO_3$. In our dataset, there was a highly significant correlation between exchangeable and





HNO$_3$-extractable Ca for a subset of soils (0-30 cm depth, r = +0.60, *P* < 0.001, *n* = 181; Fig. S5) and exchangeable Ca can be considered a representative measure for the carbonate content in the soil.

350 The pH-dependent influence of SOM-stabilizing minerals can also explain the observed differences between simulated and measured SOC stocks at regional scale (Fig. 3, Fig. S1). The largest differences, with underestimates of SOC stocks by 7 kg C m$^{-2}$, were found in the Southern Alps. The Southern Alps are characterized by high contents of Fe- and Al-oxides (Table 1) that are considered the main drivers of the large SOC storage in this region (Blaser et al., 1997). The high SOC stocks (Fig. 3) can additionally be related to the black carbon accumulated in these soils due to frequent forest fires and

355 charcoal production (Eckmeier et al., 2010). Here, we could not quantify fire-derived black carbon and its potential contribution to SOM stabilization. Even when the Southern Alps were excluded from the statistical analysis, exchangeable Fe was a significant predictor of SOC stocks (Table S4a), as well as of Yasso20 deviations (Table S4b), being the predictor explaining the largest portion of SOC stocks model variance in all regions with mostly acidic soils (Table S5). This confirms the key role of pedogenic oxides for SOM stabilization at low pH conditions.

360 **4.2 Climatic influences on SOC stocks**

 Beyond soil parameters, precipitation was a key driver of SOC stocks, with SOC stocks increasing with increasing MAP (Table 2, Fig. 4). This pattern likely resulted from a number of mechanisms: (i) a retarded decomposition at anaerobic microsites (Keiluweit et al., 2017), (ii) a reduced litter quality by an increasing contribution of conifers towards higher MAP, which also coincides with low MAT at high altitudes, as confirmed by a negative correlation of MAP with MAT for

365 elevation > 1000 m (r = -0.43, *P* < 0.001) that is typical for alpine regions (Prietzel and Christophel, 2014), (iii) a higher transfer of dissolved organic matter into the mineral soil with increasing soil water fluxes, and (iv) an enhanced weathering in moister climate that fosters mineral surface reactivity (Doetterl et al., 2015). Precipitation was also a significant driver of Yasso20 deviations (Table 3, Table S4), with MAP accounting for up to 20-25% of SOC stocks variance in the Alps and Plateau (Table S5). Yasso20 underestimated SOC stocks when MAP exceeded 1400 mm yr$^{-1}$ (Fig. S3). Consistently, Viskari

370 et al. (2022) reported that Yasso20 underestimated SOC amounts at higher precipitation, given that increasing precipitation over a certain threshold does not reduce the SOC decomposition rates in Yasso20. Current Earth System Models also tend to overestimate heterotrophic respiration fluxes where precipitation levels are high (Guenet et al., 2024). Our analysis of Yasso20 deviations, showing that SOC stocks were underestimated in waterlogged soils with macroscopic signs of anaerobic conditions (Fig. 3), confirms that anaerobic conditions at the microscale or the impeded drainage at the profile or plot scale

375 lead to a drastic decline in SOC mineralization rates (Hagedorn et al., 2000; Keiluweit et al., 2017). These conditions are not captured by the Yasso20 model, as soil moisture is currently not included as a model driver (Viskari et al., 2022), but could potentially be resolved by including soil moisture at monthly time steps, and by coupling Yasso to a soil water model (Guenet et al., 2024). Large differences between simulated and measured SOC stocks were also found in poorly drained soils of Norway and Finland using the previous version Yasso07, with soil moisture regimes overruling the importance of tree

380 productivity (Dalsgaard et al., 2016). The model deviations were additionally attributed to the high contribution of



understory vegetation in high rainfall areas (De Wit et al., 2006; Lehtonen et al., 2016). Understory including herb and shrub layers is normally not accounted in forest inventories or satellite-based NPP data (Mao et al., 2019), though it can contribute significantly to the annual litter production (Didion, 2020). In addition to its effect on the biological activity of the plant-soil system, precipitation is linked to mineral weathering, forming reactive mineral surfaces (Kramer and Chadwick, 2018). In

Swiss forest soils, MAP was indeed in the same principal component (PC3) as clay (Table S1a), positively correlating with clay ($r = +0.21$, $P < 0.001$) as well as with exchangeable Fe and Al (with Fe: $r = +0.10$, $P = 0.019$, with Al: $r = +0.14$, $P < 0.001$) which supports the idea that high MAP may indirectly influence SOC stabilization through its effects on the reactivity of mineral surfaces. At the other extreme, at low precipitations, Lehtonen et al. (2016) observed an underestimation of SOC stocks by Yasso07, which was attributed to the poor representation of drought effects on SOM decomposition. In fact,

Yasso07 (but also Yasso20) simulations are based on yearly time steps using annual precipitation, which does not capture seasonally extreme dry or moist conditions (Lehtonen et al., 2016; Viskari et al., 2022) and the uneven distribution of summer precipitations (Thürig et al., 2005) thereby yielding to misestimates of soil C stocks.

Across Swiss forests, organic layer C stocks increased with the contribution of conifers (Table S6), likely due to a slower decomposition of the more recalcitrant litter inputs (Heim and Frey, 2004) and fostered by the colder conditions at

higher elevations where conifers are more abundant. In our study, MAT did not significantly influence organic layer as well as total SOC stocks (Table 2, Table S6), possibly due to the positive correlation of MAT with the share of broadleaves ($r = +0.58$, $P < 0.001$) which did not allow to fully disentangle the specific effects of forest types and MAT. Accounting for the regional variability in the linear mixed-effect model, an overall negative effect of MAT was estimated for Yasso20 deviations (Table 3). This finding is in agreement with the tendency of Yasso20 to underestimate SOC with increasing

temperatures (Viskari et al., 2022). The Yasso model generally captures well the effects of temperature and vegetation type, as shown across regional environmental gradients in Swiss forests (Didion et al., 2014). Since Yasso20 deviations increased with temperature (Fig. S3) with Yasso20 overestimating SOC stocks in low-elevation broadleaf forests of the Swiss Plateau but not in coniferous stands at higher elevation (Fig. 3, Fig. S1), we interpret this overestimation as an indication for an impact of past land-use and current land management. The Plateau has intensively been used for agriculture since Roman

times (Haas et al., 2020), which depleted soils in SOC (Thürig et al., 2005). It is still Switzerland's most intensively managed region, with harvesting exceeding tree growth increments (Thürig et al., 2021). This intensive management, particularly harvesting, may not be adequately captured by Yasso20 simulations. On the one hand, harvesting depletes soil C stocks by changing the microclimate, reducing the litter inputs and leading to physical soil disturbances (Mayer et al., 2024). On the other hand, timber removal or other small-scale forest disturbances are poorly detected by satellite-driven NPP

estimates (Neumann et al., 2015; Park et al., 2021). Consequently, our MODIS-derived NPP inputs likely overestimated the actual litter inputs in forests with intensive harvesting. Similarly, simulations with Yasso07 in French temperate forests tended to overestimate SOC stocks in broadleaf stands while underestimating them in coniferous ones, which was attributed to historical differences in land use and stand age between broadleaf and coniferous sites (Mao et al., 2019).





### 4.3 Performance and potential improvements of Yasso20

Despite the local and regional deviations between SOC stocks simulated with Yasso20 and the measured ones, our study showed that averaged across Swiss forests and thus at the national scale, Yasso20 reproduced measured SOC stocks (Fig. 3), confirming the wide applicability of Yasso20 model (Viskari et al., 2022). The Yasso model was developed for general uses on primarily forest land, including GHG reporting at national level and requires less input information (no soil data) compared to more detailed soil C models (Liski et al., 2005). Yasso is compatible with biomass data from National Forest Inventories, where soil parameters are often not measured, allowing to estimate soil C stock changes (Hernández et al., 2017). Our study indicates that implementing SOM stabilization mechanisms would improve the Yasso model and potentially also other similar soil C models such as RothC model that accounts only for clay content (Coleman and Jenkinson, 2014). However, detailed soil information controlling SOM stabilization is often missing at larger scales. We therefore suggest to further explore commonly available soil parameters, potentially by simple pedo-transfer functions (e.g. based on exchangeable Fe and Ca) capturing SOC stabilization at different pH levels. Our study also revealed that Yasso did not capture SOC stocks (and potentially SOC stock changes) at high precipitation levels (here MAP > 1400 mm yr$^{-1}$). This hampers predictions especially under extreme or variable intra-annual precipitation patterns that will become more frequent under climate change. Obviously, linking Yasso20 to a soil water model accounting for drought and waterlogging processes would have a high potential for improving estimates of SOC stocks. However, a robust modeling of waterlogging would also require spatially highly resolved soil information.

Our study does not inform on the performance of Yasso20 in estimating SOC stock changes, which is the primary application of Yasso for GHG reporting. However, it seems likely that also SOC stock changes will be affected by soil physico-chemical properties, due to the importance of organo-mineral interactions for SOC stabilization and thus long-term C sequestration (Wiesmeier et al., 2019; Bramble et al., 2023). This would be congruent with the concept of soil C saturation suggesting that effects of C inputs on soil carbon storage depend upon inherent physico-chemical limitations (Stewart et al., 2007; Six et al., 2024).

### 5. Conclusions

Our study demonstrates that soil mineral properties controlling SOM stabilization play a dominant role for SOC stocks across Swiss forest soils. The control appears to be pH dependent with Fe (and Al) having a major influence in acidic soils, while interactions with Ca are most important in soils with pH values above 5. Not accounting for these processes in a soil C model designed to estimate national scale SOC stocks and their changes - based solely on litter inputs and climate - can lead to underestimating SOC stocks in certain regions, particularly those rich in SOM stabilizing minerals. Despite its simple model structure, our results show that Yasso20 yielded on average SOC stocks comparable to measured values at 556 forest sites across Switzerland. This supports Yasso20 broad applicability for predicting forest SOC stocks at larger national scales. Including SOM stabilization and linking soil C models to soil water models has a high potential to improve the accuracy of

the estimates. It remains uncertain, however, whether the model lack of representation of mineral-driven SOM stabilization or/and of varying soil moisture regimes will not only affect SOC stock estimates but also expected changes in SOC stocks - the primary application of Yasso in GHG inventories.

**Code and Data availability**

All data used in this study are available online through EnviDat at https://doi.org/10.16904/envidat.443 (Guidi et al., 2024). The code can be made available upon request.

**Author contributions**

FH and PN conceived and designed the study. CG and SG performed data analysis and visualization and wrote the manuscript with contributions from all co-authors. All authors contributed to the interpretation of the findings, they read and approved the submitted version.

**Competing interests**

One of the (co-)authors is a member of the editorial board of Biogeosciences.

**Acknowledgements**

This study (SNF 406840_143025) was funded by the Swiss National Fond (SNF) within the National Research Programme 68 (Sustainable Use of Soil as a Resource). Evaluations were based on data from the Swiss Long-term Forest Ecosystem Research programme LWF (www.lwf.ch), which is part of the UNECE Co-operative Programme on Assessment and Monitoring of Air Pollution Effects on Forests ICP Forests (www.icp-forests.net). We are particularly grateful to Peter Waldner for the provision of the LWF data, to Oliver Schramm for the collection of the data, and to Peter Jakob for the technical support on the database. The authors also especially acknowledge Flurin Sutter for helping with preparation of Fig. 1, Achilleas Psomas for his support with obtaining NPP data and Sophia Etzold for providing long-term tree allocation data in LWF. We also acknowledge Jürgen Zell and Sebastian Doetterl for fruitful discussions on statistical analysis of an earlier version of the manuscript.



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
