# Peer review of "Drivers of soil organic carbon from temperate to alpine forests: a model-based analysis of the Swiss forest soil inventory with Yasso20"

_EGUsphere, 2024_

## Author Comment (AC1)

**Reply to Anonymous Referee #1**

This manuscript aims to test the soil carbon model Yasso20 against a national scale dataset of soil carbon stock in Swiss forests. The paper is well written with clear objectives and generally sound methods (but see below). The core result, that is the soil mineral characteristics are primary drivers of soil carbon stock and Yasso20's predictive bias, is interesting for the whole modelling community.

**Author's response**

We thank the Referee for the constructive comments and suggestions, which helped to improve the quality and clarity of our manuscript. We have considered each of your remarks and modified the manuscript accordingly. You can find below our responses to your comments.

**Reviewer comment**

The main weakness of the manuscript comes to the lack of details about the forest context, litter and soil carbon estimation and modelling procedures:

More detailed information on forest management (LN87) and disturbance history (LN88) would be appreciated. For example, how were tree branches harvested? (Models usually ask for this kind of size thresholds that can play a role in decomposition rates). Western Europe, including Switzerland, is known to have suffered from storms in the past, including the Lothar in 1999 (see https://www.wsl.ch/en/news/25-years-after-lothar-how-the-windstorm-rebuilt-the-forest/). So none of these forest plots were attacked?

**Author's response**

Thank you for your remarks. Our study focuses on forests older than 120 years (old-growth forests, identified by the use of historic maps; Gosheva et al. (2017)). As the soil sampling was performed on undisturbed sites, none of the sampled 556 sites was impacted by the storms Vivian (in 1990) or Lothar (in 1999). Although most of the Swiss forests are regularly exploited (62% of the forests, up 79% in the Swiss Plateau), the harvesting practices normally consist of planned loggings of single trees with harvesting residues (typically the smaller branches and tree top, foliage and belowground part with trunk to approx. 30 cm aboveground) normally left in the stand (Brändli et al., 2020). This keeps the disturbance by management interventions at a minimum. We are aware that natural disturbances from windthrows (i.e. Vivian and Lothar) in Swiss forests led to SOC losses especially in the organic layers of forests at high elevation (1 to 2 kg C m$^{-2}$) (Mayer et al., 2023), but none of the sampled sites were impacted.

**We have now added this information at lines 87-88 in the manuscript:**

"*In combination with the typically small-scale forest management in Switzerland, which include planned loggings of single trees with harvesting residues (e.g. fine branches, foliage, belowground part with trunk to approx. 30 cm aboveground) normally left in the stand (Brändli et al., 2020), the focus on old forests ensures that there have been only reduced*

*disturbances in the forest cover over the past decades. Although natural disturbances such as windthrows occurred in Swiss forests (e.g. Vivian in 1990 and Lothar in 1999) with possible impact on SOC stocks (Mayer et al., 2023), none of the forest sites was affected as sampling excluded windthrow sites."*

**Reviewer comment**

More detailed information should be provided on how the NPP (LN150-1) was obtained (please avoid only citing other papers) and, especially, how it was converted into the estimate of annual litter input. Since litter input is derived from model outputs, so what is the suitability and prediction quality of the model (Terra and Aqua MODIS-satellite) for Swiss forests? This is not mentioned in LN152-153. As a result, one might ask about the reliability of the input, especially when reading LN410-411.

**Author's response**

Thank you for your comment. Since no stand inventory or litter input measurements were available at the investigated soil sites, we obtained the NPP from Terra and Aqua MODIS satellite (500-m resolution). We considered the NPP as a proxy of long-term litter C input to the soil, which was used to simulate SOC stocks at steady state with Yasso (see line 150). This is consistent with the approach of published SOC model applications (Abramoff et al., 2022; Pierson et al., 2022), and with the current Yasso20 calibration (Viskari et al., 2022), where the litter input was similarly obtained from the GPP product of Terra and Aqua MODIS satellite, setting a maximum NPP to GPP ratio to 0.5. In our study, we also considered a maximum NPP to GPP ratio of 0.5, which agrees well with the NPP to GPP ratio (0.42-0.49) observed at two contrasting forest ecosystems in Switzerland (Etzold et al., 2011).

We also tested the suitability of Terra and Aqua MODIS-satellite to estimate NPP at Swiss forests by comparing the NPP obtained by Terra and Aqua MODIS-satellite with NPP derived by stand inventory data at 18 Swiss forest sites of the long-term forest ecosystem research LWF (Etzold et al., 2014). On average for the period 2001-2010, the mean NPP estimated by the satellite approach was $0.49\pm0.04$ kg C m$^{-2}$ yr$^{-1}$, while a mean NPP of $0.46\pm0.05$ kg C m$^{-2}$ yr$^{-1}$ was estimated by the terrestrial approach for the same period (Etzold et al., 2014). In addition, the satellite NPP proves to be a reasonable proxy of large range of forest productivity across Swiss forests (see new **Fig. S1a** below) that is consistent with the gross volume increment trends across Swiss forest regions shown by the Swiss NFI (Brändli et al., 2020).

Taken together, these results confirm the suitability of the NPP obtained Terra and Aqua MODIS-satellite as a proxy of long-term litter inputs for Swiss forest conditions.

**We have now expanded and clarified in the manuscript the limitations of the approaches (satellite and terrestrial) to estimate litter inputs at line 413 of the discussion, and added Figure S1a to the supplement:**

*"Satellite-derived NPP is here used as input for Yasso simulations of SOC stocks at steady state. Since direct measurements of forest stands and detailed information of soil C inputs are*

*often lacking at larger scales - as in this study - the use of NPP as a proxy of long-term litter C input to the soil is consistent with SOC model applications at the regional and global scales (Abramoff et al., 2022; Pierson et al., 2022), as well as in the calibration of Yasso20 (Viskari et al., 2022). We are aware that litter input derived from satellites can be uncertain, thus potentially contributing to the observed discrepancies between simulated and measured SOC stocks at the site level. In fact, the fine scale variability in litter inputs cannot be captured by satellite-derived NPP estimates given (1) the larger pixel size of MODIS (500 m x 500 m) compared to the site scale of the soil sampling, and (2) the partitioning into tree components using average allocation factors, due to the lack of data at the site level. Satellite-derived NPP may have resulted in an overestimation of the litter input in regions with intensive forest management as in the Plateau, since small-scale disturbances such as thinning are not well detected by satellite estimates (Neumann et al., 2015; Park et al., 2021). Lastly, forests allocate a portion of NPP not only to fast-cycling components that are annually returned to the soil (i.e. fine roots and foliage) but also to components with slower turnover time such as stems and branches. Nevertheless, the satellite NPP approach proves to be a reasonable proxy of the large range of forest productivity across Swiss forests, i.e. ranging from 0.3 kg C $m^{-2}$ $yr^{-1}$ in the Alps to 0.8 kg C $m^{-2}$ $yr^{-1}$ at the warmest sites (see **Fig. S1a**), which is consistent with the gross volume increment shown across different regions in the Swiss NFI (Brändli et al., 2020). Moreover, at the 18 sites of the long-term forest monitoring program LWF, the mean NPP over the period 2001-2010 based on the satellite approach amounted to $0.49\pm0.04$ kg C $m^{-2}$ $yr^{-1}$ as compared to $0.46\pm0.05$ kg C $m^{-2}$ $yr^{-1}$ estimated by the terrestrial approach for the same period (Etzold et al., 2014).*

*Similarly, terrestrial methods based on forest inventories may lead to uncertain estimates of litter inputs. These uncertainties mostly relate to (1) country-specific allometries and expansion factors used to estimate tree biomass, (2) turnover times applied to obtain the litter inputs, and (3) failing to appropriately estimate inputs from fine roots and understory vegetation, which remain severely unconstrained despite their major contribution to forest soil C inputs (Didion, 2020; Neumann et al., 2020)."*

[Figure]

**Fig. S1a**. *Net primary production (NPP) across Swiss forest regions, excluding waterlogged soils. Total n sites = 468. Letters indicate significantly different across regions, based on ANOVA followed by Tukey's test with P < 0.05.*

**Reviewer comment**

Then, how was tree litter chemical quality split? It is not very clear in the text (LN158-160). Was it partitioned at the level of tree species or at the level of tree functional type (broadleaved versus coniferous)?

Also, so a fixed set of allocation factors were used to estimate tree component (LN155) without distinguishing between tree functional type?

**Author's response**

We appreciate your remarks and suggestions. The litter chemical quality was split into AWEN fractions based on the percentage of broadleaf and conifer species at each site recorded by field assessments (line 115-118). First "*we partitioned the NPP into broadleaf and conifer species, multiplying the NPP by the percentage of broadleaf and conifer species recorded by field assessments at each site*" (line 153). Then the NPP – after being partitioned into broadleaf-derived and conifer-derived NPP - was split into AWEN fractions using the functions and approach used in the Swiss GHG inventory (Didion, 2023) based on measured fractions in the long-term forest monitoring program LWF (Didion et al., 2014).

**We have now clarified the methodology used at lines 155 and 158-160 in the manuscript:**

"*The C inputs for each pool were separated into the four AWEN components based on the percentage of broadleaf and conifer species at each site, using the functions and approach adopted in the Swiss GHG inventory (Didion, 2023) based on measured fractions at sites of the long-term forest monitoring program LWF (Didion et al., 2014).*"

In this study, the estimates of NPP allocation into tree components rely on long-term forest ecosystem data from LWF network (Etzold et al., 2014), see lines 154-157. We did not distinguish between tree species or tree functional types, since we did not observe major differences between different tree functional types in C allocation to different tree compartments. In the future, this approach can be further refined by establishing species-specific C allocation factors, which should also consider the effect of climatic conditions on C allocation.

**Reviewer comment**

I find the categorization (LN119) risky. In this case, stands of 40-49% broadleaf and stands of 51-60% broadleaf which are actually quite similar stands in term of species mixture, will belong to two contrasting categories. I wonder if a new category of "mixed stand" should be proposed and compared with more pure stands? Also, how was the chemical quality of the litter in these mixed stands partitioned? As a reader, I am also curious to know how big a difference that the effect of species functional type can theoretically make in Yasso20 simulations in your case? (According to Fig. 3b, there is limited difference in the model's predicts). Would it be possible to do an uncertainty analysis on how the 0%, 50% and 100% of broadleaves at a given site (with given climatic inputs) can change the predictions? This can be done for all sites or only for those where both broadleaves and conifers coexist,

enabling us to better understand the sensitivity of the species related parameters used in your case to the model predictions.

**Author's response**

Thank you for your suggestions. In the manuscript, we have separated forest sites into broadleaf and coniferous forests only for visualization purposes in the Figures (e.g. Figure 2-4). The percentage of broadleaf species recorded at each site has already been considered for partitioning NPP input at each site, which is then used as model input for Yasso20 simulations. We have specified this at line 152-154: "*We partitioned the NPP into broadleaf and conifer species, multiplying the NPP by the percentage of broadleaf and conifer species recorded by field assessments at each site*".

**We have now reformulated the sentence at line 118-119:**

"*Only for visualization purposes, the forest sites were subdivided into two types based on the broadleaf percentage: coniferous (0-50%) and broadleaf (51-100%) forests.*"

**Reviewer comment**

Finally, in terms of data representation and visualization, the authors chose to present simulated versus observed C using bar plots (Fig. 3), box plots (Fig. S1) or scatter plots (deviation versus factors as in Fig. S3). While it is good, it would be more informative to directly show the scatter plots of simulated C versus observed C at the site level with a 1:1 line to see if and how the model captures the observations according to different tree functional types (and/or regions). A new (or supplementary) plot at the site level would be desirable as a complement to Fig. 3.

**Author's response**

We presented the simulated vs observed C stocks using bar plots to better visualize in which regions the simulated SOC stocks deviate more from the measured stocks, in order to infer on the main processes driving mineral stabilization.

**We have now added the 1:1 plot of simulated vs observed C stocks in the supplemental material (see Fig. S2 below).** As expected, the Yasso20-simulated values show only a smaller variability in SOC stocks compared to the measured SOC across sites, although the average simulated SOC stocks match well the average SOC stocks across Swiss forest soils. Possible reasons for this are: (1) the uncertain litter input estimation, since satellite-derived NPP was used as a proxy of site productivity, (2) high spatial variability of soil properties which drives very high SOC stocks at some sites (e.g. high Fe and Al), and (3) anaerobic conditions at the site scale that retard the SOC mineralization rates and lead to locally higher SOC stocks.

[Figure]

*Fig. S2. Comparison between Yasso20-simulated and measured SOC stocks by forest types.*
*Total n sites = 468, excluding waterlogged soils.*

**Specific points:**

**Reviewer comment**

LN88: "minimal disturbances" was mentioned, but latter we read "frequent fires" (LN354). Fires are an important disturbance in the forest system. Is this a contradiction? What about storms (see above)?

**Author's response**

Thank you for the comment. We have included in the manuscript your remark that disturbances such as forest management or windthrows may affect soil C in Swiss forests at line 88 (*see previous Author's response*), although none of the sampled site was directly affected by windthrows. In Switzerland, forest fires - including wildfires during the dry season and deliberate fires common especially in the Southern Alps for over 10,000 years – contributed to increase SOC storage due to increased resistance of charred material to decomposition (Eckmeier et al., 2010). High amounts of charred material persisted in Fe- and Al-rich soils for millennia (Eckmeier et al., 2010). Here, we cannot exclude that the sampled sites have not been affected by forest fires over millennia time scales. Thus, we already discussed in the manuscript at line 353-358 how fire-derived black C may potentially affect soil C stock measured in the Southern Alps, showing that the main results remained unchanged even by excluding this region from the analysis.

**Reviewer comment**

LN125: how were the percentages of C content of the litter measured?

**Author's response**

The C content of ground organic materials (including the following horizons: undecomposed litter L, fermentation F, and humified H) and mineral soil were measured by dry combustion using an elemental analyzer (NC 2500, CE Instruments, Italy). This is reported at line 105-106.

**To improve clarity, we have added in the manuscript at line 125 the following sentence**:

"*multiplied by the percentage of C content obtained by dry combustion (see section 2.2).*"

**Reviewer comment**

LN129: it would be good to explain how the volumetric stone content was estimated.

**Author's response**

Thank you for the comment. **We have added this information in the manuscript at line 129**:

*"volumetric stone content visually estimated from the soil profile (Richard et al., 1978)"*

**Reviewer comment**

LN183: also with a standard deviation of 1 ?

**Author's response**

We have centered all independent variables before analysis to a mean of 0. We did not scale the variables to standard deviation of 1. This allows to provide in Table 2 and 3 (column Est.) meaningful model estimates, which can be interpreted in the same measurement units of the predictor and response variables.

**We have replaced the word "standardized" with "centered" in the sentence at line 183 to improve clarity:**

"*All explanatory variables were centered to a mean of 0*"

**Reviewer comment**

LN 320: very interesting result for clay. Have you also tried the sand content (which does not necessarily correlate with the clay content)?

**Author's response**

We tested whether the sand content would be significantly correlated to the SOC stocks, which we plot here below for your information. No significant correlation was found (where r is the Person correlation coefficient), similarly to the clay content (Fig. 4 in the MS):

[Figure]

In addition, a strong correlation of sand and clay content was observed in this study:

[Figure]

For these reasons, we have decided to include only the clay content (%) in the analysis presented in the manuscript.

**Reviewer comment**

Table 2 (and also other Tables that summarize statistics), I suggest using a MAP unit of 100 mm rather than mm to give a more meaningful sense for each increment of 1 unit.

**Author's response**

Thank you for your suggestion. **We will modify the Tables using a MAP unit of 100 mm instead of 1 mm.**

**Reviewer comment**

We also often see the important factor of "slope", but it is rarely mentioned. And why is that?

**Author's response**

We agree that the slope can be an important driver of SOC stocks. In fact, we have included slope as a predictor in our statistical analysis. In the MS at line 272-274, we report that slope has a negative effect on SOC stocks. To not increase further the length of our manuscript and

considering that slope did not appear a dominant driver of SOC stocks in the different biogeographic regions (Table S5), we do not discuss about slope in more detail. However, topography represents an important factor influencing SOC stocks, with soil erosion occurring on steep slopes while waterlogging in flat areas (Fernández-Romero et al., 2014) which can strongly impact the variability of SOC stocks at the site-scale.

**Reviewer comment**

Figure 2: Why was a polynomial regression also used? Would it be possible to plot the litter input data in addition to NPP?

**Author's response**

We have shown a polynomial regression in Figure 2a (in addition to the linear correlation), since this better captures the productivity trend across elevation with NPP peaking at an elevation of about 700 m a.s.l., which then decreased with increasing elevation (line 210).

Since we do not have the litter input at the sites, we considered the NPP as a proxy of long-term litter carbon input to the soil, see previous *Author's response*. We have now plotted the NPP data in the new **Fig. S1a**, which show the variability in NPP across Swiss forest regions.

**Reviewer comment**

Figure 3: I have the impression that the Yasso20 estimates for all five regions are not very different and possibly not very significant in most regions. In (b) this may also be the case: does Yasso20 predict significantly different stocks between broadleaved and coniferous stands? It would also be interesting to read statistical diagnoses somewhere among the simulated (or observed) groups.

**Author's response**

We have now added different letters to the updated **Fig. 3a,b** (see below) which indicate significantly different means (1) among the measured SOC stocks (capital letters), and (2) among the Yasso20-simulated SOC stocks (lowercase letters).

Here, we show that Yasso20-simulated SOC stocks are statistically different across regions (**Fig. 3a**) and between forest types (see **Fig. 3b**), but differences are much smaller compared to differences observed for measured SOC stocks. Yasso20-simulated SOC stocks across regions show a different pattern compared to observed SOC stocks, with Jura and Pre-Alps > Southern Alps. We regard the smaller variation of SOC stocks modelled by Yasso20 as an indication for the lacking representation of region-specific soil characteristics such as mineralogy.

**We have modified Fig. 3a,b and its caption, which now includes also the comparison among measured (capital letters), and among simulated SOC stocks (lower case letters):**

[Figure]

**Figure 3: Comparison between measured SOC stocks vs Yasso20-simulated stocks by: (a) biogeographic regions of Switzerland (waterlogged soils shown separately), and (b) forest types, excluding waterlogged soils. The measured stocks are shown as organic layers and mineral soils to 100 cm depth (total *n* sites = 556; excluding waterlogged soils, *n* sites = 468). The simulated stocks at each site are based on the mean of 500 replicate simulations representing model parameters uncertainty. SOC stocks are represented as means ± standard errors. *P*-values are calculated with Welch's *t*-tests: $P \geq 0.05$ (n.s., not significant), $P < 0.001$ (***). Capital letters indicate significantly different means among measured SOC stocks, while lowercase letters indicate significantly different means among Yasso20-simulated SOC stocks, based on ANOVA followed by Tukey's test (Figure 3a) and Welch's t-tests (Figure 3b) with $P < 0.05$.**

**Reviewer comment**

Fig. 4: It would be good to briefly explain why such a subjective cut-off point of pH = 5 was chosen for data analysis. It is particularly useful for non-soil experts, although it is intuitively logical for soil experts.

**Author's response**

Thank you for the remark. **This clarification will be added to the revised manuscript at line 254-255:**

"*We separated the dataset into soils with pH ≤ 5 and with pH > 5, since pH is recognized as a predictor of SOC stabilization mechanisms, mediated by Al- or Fe- under acidic conditions while Ca exerts a dominant influence under increasing pH (Rowley et al., 2018).*"

**References - Reply to Anonymous Referee #1**

Abramoff, R.Z., Guenet, B., Zhang, H., Georgiou, K., Xu, X., Rossel, R.A.V., Yuan, W., Ciais, P., 2022. Improved global-scale predictions of soil carbon stocks with Millennial Version 2. Soil Biology and Biochemistry 164, 108466.

Brändli, U.-B., Abegg, M., Allgaier, B.L., 2020. Schweizerisches Landesforstinventar: Ergebnisse der vierten Erhebung 2009-2017. WSL.

Didion, M., 2020. Extending harmonized national forest inventory herb layer vegetation cover observations to derive comprehensive biomass estimates. Forest Ecosystems 7, 1-14.

Didion, M., 2023. Data on soil carbon stock change, carbon stock and stock change in surface litter and in coarse dead wood prepared for the Swiss NIR 2024 (GHGI 1990–2022).

Didion, M., Frey, B., Rogiers, N., Thürig, E., 2014. Validating tree litter decomposition in the Yasso07 carbon model. Ecological modelling 291, 58-68.

Eckmeier, E., Egli, M., Schmidt, M., Schlumpf, N., Nötzli, M., Minikus-Stary, N., Hagedorn, F., 2010. Preservation of fire-derived carbon compounds and sorptive stabilisation promote the accumulation of organic matter in black soils of the Southern Alps. Geoderma 159, 147-155.

Etzold, S., Ruehr, N.K., Zweifel, R., Dobbertin, M., Zingg, A., Pluess, P., Häsler, R., Eugster, W., Buchmann, N., 2011. The carbon balance of two contrasting mountain forest ecosystems in Switzerland: similar annual trends, but seasonal differences. Ecosystems 14, 1289-1309.

Etzold, S., Waldner, P., Thimonier, A., Schmitt, M., Dobbertin, M., 2014. Tree growth in Swiss forests between 1995 and 2010 in relation to climate and stand conditions: Recent disturbances matter. Forest Ecology and Management 311, 41-55.

Gosheva, S., Walthert, L., Niklaus, P.A., Zimmermann, S., Gimmi, U., Hagedorn, F., 2017. Reconstruction of historic forest cover changes indicates minor effects on carbon stocks in Swiss forest soils. Ecosystems 20, 1512-1528.

Mayer, M., Rusch, S., Didion, M., Baltensweiler, A., Walthert, L., Ranft, F., Rigling, A., Zimmermann, S., Hagedorn, F., 2023. Elevation dependent response of soil organic carbon stocks to forest windthrow. Science of the Total Environment 857, 159694.

Neumann, M., Godbold, D.L., Hirano, Y., Finér, L., 2020. Improving models of fine root carbon stocks and fluxes in European forests. Journal of Ecology 108, 496-514.

Neumann, M., Zhao, M., Kindermann, G., Hasenauer, H., 2015. Comparing MODIS net primary production estimates with terrestrial national forest inventory data in Austria. Remote Sensing 7, 3878-3906.

Park, J.H., Gan, J., Park, C., 2021. Discrepancies between global forest net primary productivity estimates derived from MODIS and forest inventory data and underlying factors. Remote Sensing 13, 1441.

Pierson, D., Lohse, K.A., Wieder, W.R., Patton, N.R., Facer, J., de Graaff, M.-A., Georgiou, K., Seyfried, M.S., Flerchinger, G., Will, R., 2022. Optimizing process-based models to predict current and future soil organic carbon stocks at high-resolution. Scientific Reports 12, 10824.

Richard, F., Lüscher, P., Strobel, T., 1978. Physikalische Eigenschaften von Böden der Schweiz. Eidgenössischen Anstalt für das forstliche Versuchswesen.

Rowley, M.C., Grand, S., Verrecchia, É.P., 2018. Calcium-mediated stabilisation of soil organic carbon. Biogeochemistry 137, 27-49.

Viskari, T., Pusa, J., Fer, I., Repo, A., Vira, J., Liski, J., 2022. Calibrating the soil organic carbon model Yasso20 with multiple datasets. Geoscientific Model Development 15, 1735-1752.

---

## Author Comment (AC2)

**Reply to Anonymous Referee #2**

Review of" Drivers of soil organic carbon from temperate to alpine forests: a model-based analysis of the Swiss forest soil inventory with Yasso20".

Authors analyzed significance of soil properties especially missing organo-mineral (OM) interactions on mismatch between the soil organic C (SOC) stocks measured in Alps and their modeled counterparts estimated with Yasso20 model. The study itself is interesting and relevant.

The conclusion is in line with the expectations of the theoretical set up of the analysis. Authors conclude that the model failed to estimate SOCs in regions where OM is relevant mechanism of total SOC accumulation and where dependency of decomposition on precipitation could not capture its reduction related to higher soil moisture levels. Authors thus demonstrate that the model indeed fails to estimate precisely the SOC variability and conclude that if the model is applied in Alps, then it needs further development. This is known in literature and not new.

What is new is nicely demonstrating where the model **(applied as in this study)** fails in Alps and how much it fails. Simplified, these are the soils with ph <5 showing Fe correlation to SOC and ph > 5 showing Ca correlation to SOC, and soils with MAP > 1400 mm.

**Author's response**

We thank the Anonymous Referee 2 for the constructive comments and suggestions, which helped to improve the quality and clarity of our manuscript. We have considered each of your remarks and modified the manuscript accordingly. You can find below our responses to all your comments.

**Reviewer comment**

However, authors also demonstrate that the SOC discrepancies between measurements and Yasso20 could be estimated more precisely with statistical models (explaining about 50% of variance). Although, how much of total variance is explained in combination of Yasso + statistical models is not known. The final conclusions revolve about findings expected from the analysis setup. The motivation why SOC needs to be evaluated in combination of statistical and process models is not clear.

**Author's response**

Thank you very much for your suggestion. In this study, we make use of the deviations between Yasso20-simulated SOC stocks and measured SOC stocks to more clearly disentangle the main drivers of SOC stocks across an extensive Swiss forest soils dataset. Since Yasso is a simple soil C model driven by litter inputs and climate, deviations between simulated and measured SOC stocks can be mostly related to drivers of SOC stocks not accounted in the models, such as mineral soil properties, which are not explicitly implemented in Yasso. For this reason, we have sequentially analyzed the Yasso deviations by statistical model, which allows us to evaluate the effects of different predictors on Yasso deviations.

Although a combined approach – directly coupling Yasso with a statistical model - would be very promising since it will allow to incorporate predictors that are not explicitly included in the Yasso model (e.g. mineral soil properties), this would be beyond the focus of this manuscript. As a first step in this direction, we here demonstrate what are the: (1) main drivers of SOC stocks, and (2) main drivers of Yasso20 deviations for the Swiss forest soils. This could contribute to further improve Yasso in future works.

**We have now clarified the rationale for using Yasso followed by statistical analysis at line 69-72 in the introduction**:

*"Here, we aimed to identify the main factors controlling SOC stocks in Swiss forest soils across a large gradient of climate, soil biogeochemistry and forest types. To disentangle the main drivers of SOC stocks, our main approach was to (1) simulate SOC stocks in forest soils by Yasso20, driven only by litter input and climate, and (2) statistically analyze the deviations between Yasso20-simulated and measured SOC stocks. This allows to evaluate the importance of mineral-driven SOC stabilization, since mineral soil properties are not explicitly implemented in Yasso."*

**We have added at line 426 in the discussion the suggestion of a combined approach of Yasso and statistical model:**

"*A combined approach – directly coupling Yasso with a statistical model - would allow to account for additional parameters (as mineral soil properties) that are currently not included as model drivers but are known to be important factors controlling SOC stabilization.*"

**Reviewer comment**

All mismatch between Yasso and measurements of SOC is blamed on Yasso missing relevant processes in the model structure of decomposition, though the litter C input itself which is the most essential driver of Yasso modeled SOC was estimated using rather simple imprecise approach with high level of uncertainty. **Given that the C inputs were estimated with low precision (litter inputs C quantity represented average for all species, litter input C quality differing only between conifers and broadleaves, and C input representing rather large spatial unit of 0.25 km2 derived from last 20 years compared to soil samples representing much smaller unknown unit cc 10 m2 representing SOC development changes of 1000+ years?) this should be taken into consideration when interpreting the modeled SOC data. At present state, the study is biased by ignoring impact of imprecise estimates of C input data used to run Yasso.**

**Author's response**

Thank you very much for your remark. We are aware that NPP represents an approximation of the productivity at each site and a proxy of long-term C input to the soil. Unfortunately, no stand inventory or litter input measurements were available at the sampled sites. Nonetheless, our approach to use NPP (here derived from Terra and Aqua MODIS satellite 500-m resolution) as input of the soil C model Yasso (see line 150) is consistent with the approach of

published SOC model applications (Abramoff et al., 2022; Pierson et al., 2022), and with the current Yasso20 calibration (Viskari et al., 2022), where NPP is used as model input.

Considering that NPP represents an approximation of long-term litter C input to the soil, we address the limitations of this satellite approach in the updated manuscript. In addition, we discuss that terrestrial-based methods may also lead to biased estimates of litter inputs, e.g. due to the application of country-specific allometric equations or due to underestimates of C inputs from rhizosphere or understory vegetation. Both satellite- and terrestrial-based approaches to estimate litter inputs are normally obtained from the last few decades of measurements, while SOC turns over on decadal to millennia timescales.

Despite the difficulties with any approach in estimating C inputs, we are certain that the large-scale pattern in C inputs across Swiss forests with MATs ranging between 2 and 12°C and MAPs ranging between 400 and 2200 mm is robust. Therefore, also the magnitude of the differences between regions explored here appears robust.

**We have now expanded and clarified the limitations of litter input estimates at line 413 in the discussion, and added Figure S1a to the supplement:**

*"Satellite-derived NPP is here used as input for Yasso simulations of SOC stocks at steady state. Since direct measurements of forest stands and detailed information of soil C inputs are often lacking at larger scales - as in this study - the use of NPP as a proxy of long-term litter C input to the soil is consistent with SOC model applications at the regional and global scales (Abramoff et al., 2022; Pierson et al., 2022), as well as in the calibration of Yasso20 (Viskari et al., 2022). We are aware that litter input derived from satellites can be uncertain, thus potentially contributing to the observed discrepancies between simulated and measured SOC stocks at the site level. In fact, the fine scale variability in litter inputs cannot be captured by satellite-derived NPP estimates given (1) the larger pixel size of MODIS (500 m x 500 m) compared to the site scale of the soil sampling, and (2) the partitioning into tree components using average allocation factors, due to the lack of data at the site level. Satellite-derived NPP may have resulted in an overestimation of the litter input in regions with intensive forest management as in the Plateau, since small-scale disturbances such as thinning are not well detected by satellite estimates (Neumann et al., 2015; Park et al., 2021). Lastly, forests allocate a portion of NPP not only to fast-cycling components that are annually returned to the soil (i.e. fine roots and foliage) but also to components with slower turnover time such as stems and branches. Nevertheless, the satellite NPP approach proves to be a reasonable proxy of the large range of forest productivity across Swiss forests, i.e. ranging from 0.3 kg C $m^{-2}$ $yr^{-1}$ in the Alps to 0.8 kg C $m^{-2}$ $yr^{-1}$ at the warmest sites (see Fig. S1a), which is consistent with the gross volume increment shown across different regions in the Swiss NFI (Brändli et al., 2020). Moreover, at the 18 sites of the long-term forest monitoring program LWF, the mean NPP over the period 2001-2010 based on the satellite approach amounted to 0.49±0.04 kg C $m^{-2}$ $yr^{-1}$ as compared to 0.46±0.05 kg C $m^{-2}$ $yr^{-1}$ estimated by the terrestrial approach for the same period (Etzold et al., 2014).*

*Similarly, terrestrial methods based on forest inventories may lead to uncertain estimates of litter inputs. These uncertainties mostly relate to (1) country-specific allometries and expansion factors used to estimate tree biomass, (2) turnover times applied to obtain the litter inputs, and (3) failing to appropriately estimate inputs from fine roots and understory vegetation, which remain severely unconstrained despite their major contribution to forest soil C inputs (Didion, 2020; Neumann et al., 2020)."*

[Figure]

***Fig. S1a****. Net primary production (NPP) across Swiss forest regions, excluding waterlogged soils. Total n sites = 468. Letters indicate significantly different across regions, based on ANOVA followed by Tukey's test with P < 0.05.*

**Reviewer comment**

From the study it is difficult to understand whether it makes sense at all to use Yasso model. Although, if Yasso's decomposing of fraction of NPP explains anything than it is somewhat hinted that the model combination would yield better results than statistical models only. The main strength of combination of Yasso and statistical models over only statistical models is, however, the potential for modelling not only SOC stocks but also their changes over time (needed for estimating of ecosystem C fluxes in warming climates).

**Author's response**

To evaluate the effects of the main drivers of SOC stocks, our approach was to analyze Yasso deviations from measured SOC stocks using a statistical model. An improvement of Yasso may represent the next step. Despite the regional differences, Yasso20 captured the average SOC stocks for Switzerland. Please also refer to our *Author's response* above. As reported in the manuscript at line 432-433, we do not simulate here SOC stock changes, since no repeated SOC stock measurements were available at the sites.

**In addition to the clarification in the introduction regarding the rationale for using Yasso followed by statistical analysis (at line 69-72, see above), we also added a sentence to facilitate result interpretation at line 176:**

*"Yasso20 deviations (simulated minus measured values of total SOC stocks) were tested using linear mixed-effect models with the nlme package, version 3.1–160 (Pinheiro et al., 2022) to assess effects of the main drivers of SOC stocks on model discrepancies".*

**Reviewer comment**

In theory in ideal application of Yasso, the litter input C quantity and quality should reflect implicitly the site environmental conditions including the soil properties. However, from the analysis here it could be expected that litter inputs represented only the average for the region and thus the SOC estimates also only represent the average and could not be expected to show the small-scale variability. For example, it can be expected that if you plot derived C inputs on 2$^{nd}$ axis in Fig.3 these would be only similar averages for different regions with not much variation as modeled SOCs, unlike variation seen in measured SOCs.

**Author's response**

While we agree with the reviewer that Yasso ideally captures litter inputs driven by environmental variables, it cannot capture all effects of soil properties on SOC stocks that lead to SOC stabilization in the soil (mineralogy) or suppress decomposition (anaerobic conditions) as they are not explicitly included in the model.

We are also aware that the fine scale variability in litter input cannot be fully detected by satellite-derived NPP estimates given the larger pixel size of MODIS (500 m x 500 m) (see *Author's response* above, and the changes in the manuscript already reported above). However, the satellite NPP approach still provides a proxy of forest productivity across Swiss forests, being able to show a large productivity range across Switzerland and statistical differences across Swiss regions (see added **Fig. S1a**), which is consistent with the productivity trend across Swiss forest regions (Brändli et al., 2020). Given the low temperatures occurring in the least productive region (i.e. the Alps) – leading to slower SOC decomposition – and the high temperature at the most productive regions – leading to faster SOC decomposition – differences in simulated SOC stocks across regions were rather limited (see added **Fig. S1b**).

**We have added a supplemental figure which show the differences in NPP across regions (see new Fig. S1a, b below):**

[Figure]

**Fig. S1**. *Net primary production (NPP) (a), and Yasso20-simulated SOC stocks (b) across Swiss forest regions, excluding waterlogged soils. Total n sites = 468. Letters indicate significantly different across regions, based on ANOVA followed by Tukey's test with P < 0.05.*

**Reviewer comment**

However, the main message is correct, as it demonstrates that Yasso model structure needs development for application in Alp and thus should account for interaction with minerals and limitation of decomposition with excessive moisture. Discussion claims that Yasso20 also fails due to missing moisture limitation (which is known) but does not notice that dependency on moisture was evaluated with Yasso07 (Tupek et al., 2024) and could have been already applied here instead of precipitation.

**Author's response**

Thank you very much for your suggestion of this promising approach to include soil moisture as model driver in Yasso, which can be especially useful for peatland forest C modelling (Ťupek et al., 2024). Given that the moisture modifier of Yasso07 was developed specifically for nine sites in a boreal forest–mire ecotone in Finland, the applicability of this approach would require more tests in a larger number of experimental study sites covering Swiss forests, before being applied to this large Swiss forest soil dataset. This would go beyond the aim of this paper.

**We have taken your remark into account in the manuscript at line 376-378:**

"*this could be resolved in the future by including soil moisture at monthly time steps as model driver, or by applying a moisture modifier as shown in a boreal forest–mire ecotone in Finland for Yasso07 (Ťupek et al., 2024), or by coupling Yasso to a soil water model (Guenet et al., 2024).*"

**Reviewer comment**

My main concern is that the relation of modeled SOCs on poor litter input estimates should be elaborated in the paper to separate the mismatch in measured and modeled SOCs due to Yasso

model structure (misrepresented decomposition rates) and due to misrepresented C inputs. Showing that NPP did not correlate with SOCs (Tabe 2, and Fig. 3) does not help with confidence in estimated C inputs (which were not shown). Although, I recognize enormous work done and appreciate that the data used in the analysis was made openly available, the lack of confidence in litter input is the main weakness that requires clarifying in major revision before the paper can be accepted.

**Author's response**

Thank you very much for your comment. We have now elaborated in the manuscript that the discrepancies in the model residuals (simulated – measured SOC stocks) can also be related to uncertain litter input estimates (see *Author's response above*), in addition to the lack of implementation of soil properties and waterlogging processes in the current model formulation. Currently, we cannot fully disentangle the two components of uncertainties e.g. from litter input estimates and lack of soil processes representation in the model, but we have considered the uncertainty derived from litter input estimation by adding an additional paragraph to the manuscript.

**We have now expanded and clarified the limitations of litter input estimates at line 413 in the discussion, and added Figure S1a to the supplement:**

*"Satellite-derived NPP is here used as input for Yasso simulations of SOC stocks at steady state. Since direct measurements of forest stands and detailed information of soil C inputs are often lacking at larger scales - as in this study - the use of NPP as a proxy of long-term litter C input to the soil is consistent with SOC model applications at the regional and global scales (Abramoff et al., 2022; Pierson et al., 2022), as well as in the calibration of Yasso20 (Viskari et al., 2022). We are aware that litter input derived from satellites can be uncertain, thus potentially contributing to the observed discrepancies between simulated and measured SOC stocks at the site level. In fact, the fine scale variability in litter inputs cannot be captured by satellite-derived NPP estimates given (1) the larger pixel size of MODIS (500 m x 500 m) compared to the site scale of the soil sampling, and (2) the partitioning into tree components using average allocation factors, due to the lack of data at the site level. Satellite-derived NPP may have resulted in an overestimation of the litter input in regions with intensive forest management as in the Plateau, since small-scale disturbances such as thinning are not well detected by satellite estimates (Neumann et al., 2015; Park et al., 2021). Lastly, forests allocate a portion of NPP not only to fast-cycling components that are annually returned to the soil (i.e. fine roots and foliage) but also to components with slower turnover time such as stems and branches. Nevertheless, the satellite NPP approach proves to be a reasonable proxy of the large range of forest productivity across Swiss forests, i.e. ranging from 0.3 kg C m$^{-2}$ yr$^{-1}$ in the Alps to 0.8 kg C m$^{-2}$ yr$^{-1}$ at the warmest sites (see **Fig. S1**), which is consistent with the gross volume increment shown across different regions in the Swiss NFI (Brändli et al., 2020). Moreover, at the 18 sites of the long-term forest monitoring program LWF, the mean NPP over the period 2001-2010 based on the satellite approach amounted to 0.49±0.04 kg C m$^{-2}$ yr$^{-1}$ as compared to 0.46±0.05 kg C m$^{-2}$ yr$^{-1}$ estimated by the terrestrial approach for the same period (Etzold et al., 2014).*

*Similarly, terrestrial methods based on forest inventories may lead to uncertain estimates of litter inputs. These uncertainties mostly relate to (1) country-specific allometries and expansion factors used to estimate tree biomass, (2) turnover times applied to obtain the litter inputs, and (3) failing to appropriately estimate inputs from fine roots and understory vegetation, which remain severely unconstrained despite their major contribution to forest soil C inputs (Didion, 2020; Neumann et al., 2020)."*

[Figure]

***Fig. S1a***. *Net primary production (NPP) across Swiss forest regions, excluding waterlogged soils. Total n sites = 468. Letters indicate significantly different across regions, based on ANOVA followed by Tukey's test with P < 0.05.*

References:

Ťupek, B., Lehtonen, A., Yurova, A., Abramoff, R., Guenet, B., Bruni, E., Launiainen, S., Peltoniemi, M., Hashimoto, S., Tian, X., Heikkinen, J., Minkkinen, K., and Mäkipää, R.: Modelling boreal forest's mineral soil and peat C dynamics with the Yasso07 model coupled with the Ricker moisture modifier, Geosci. Model Dev., 17, 5349–5367, https://doi.org/10.5194/gmd-17-5349-2024, 2024.

**References - Reply to Anonymous Referee #2**

Abramoff, R.Z., Guenet, B., Zhang, H., Georgiou, K., Xu, X., Rossel, R.A.V., Yuan, W., Ciais, P., 2022. Improved global-scale predictions of soil carbon stocks with Millennial Version 2. Soil Biology and Biochemistry 164, 108466.

Brändli, U.-B., Abegg, M., Allgaier, B.L., 2020. Schweizerisches Landesforstinventar: Ergebnisse der vierten Erhebung 2009-2017. WSL.

Didion, M., 2020. Extending harmonized national forest inventory herb layer vegetation cover observations to derive comprehensive biomass estimates. Forest Ecosystems 7, 1-14.

Guenet, B., Orliac, J., Cécillon, L., Torres, O., Sereni, L., Martin, P.A., Barré, P., Bopp, L., 2024. Spatial biases reduce the ability of Earth system models to simulate soil heterotrophic respiration fluxes. Biogeosciences 21, 657-669.

Neumann, M., Godbold, D.L., Hirano, Y., Finér, L., 2020. Improving models of fine root carbon stocks and fluxes in European forests. Journal of Ecology 108, 496-514.

Neumann, M., Zhao, M., Kindermann, G., Hasenauer, H., 2015. Comparing MODIS net primary production estimates with terrestrial national forest inventory data in Austria. Remote Sensing 7, 3878-3906.

Park, J.H., Gan, J., Park, C., 2021. Discrepancies between global forest net primary productivity estimates derived from MODIS and forest inventory data and underlying factors. Remote Sensing 13, 1441.

Pierson, D., Lohse, K.A., Wieder, W.R., Patton, N.R., Facer, J., de Graaff, M.-A., Georgiou, K., Seyfried, M.S., Flerchinger, G., Will, R., 2022. Optimizing process-based models to predict current and future soil organic carbon stocks at high-resolution. Scientific Reports 12, 10824.

Pinheiro, J., Bates, D., R Core Team, 2022. nlme: Linear and Nonlinear Mixed Effects Models. R package version 3.1-160, https://CRAN.R-project.org/package=nlme.

Ťupek, B., Lehtonen, A., Yurova, A., Abramoff, R., Guenet, B., Bruni, E., Launiainen, S., Peltoniemi, M., Hashimoto, S., Tian, X., 2024. Modelling boreal forest's mineral soil and peat C dynamics with the Yasso07 model coupled with the Ricker moisture modifier. Geoscientific Model Development 17, 5349-5367.

Viskari, T., Pusa, J., Fer, I., Repo, A., Vira, J., Liski, J., 2022. Calibrating the soil organic carbon model Yasso20 with multiple datasets. Geoscientific Model Development 15, 1735-1752.

---

## Author Response (AR1)

**Author's response - *"Drivers of soil organic carbon from temperate to alpine forests: a model-based analysis of the Swiss forest soil inventory with Yasso20"**

We would like to thank the Editor and the reviewers for the time they invested on the manuscript and for their constructive comments and suggestions, which helped to improve the quality and clarity of our manuscript. You can find below our responses to all your comments. The manuscript has been revised accordingly (see *Manuscript with track changes*).

According to the editorial requests, we have also updated the color schemes of Fig. 2, 3, 4, S3 (old S1), S5 (old S3), to allow readers with color vision deficiencies to correctly interpret our findings.

To improve consistency with the definition of regions in the Swiss National Forest Inventory (NFI), we have added the reference to Fischer and Traub (2019), see line 96, and referred throughout the text to "regions" instead of "biogeographic regions".

**Anonymous Referee #1, 07 Feb 2025**

This manuscript aims to test the soil carbon model Yasso20 against a national scale dataset of soil carbon stock in Swiss forests. The paper is well written with clear objectives and generally sound methods (but see below). The core result, that is the soil mineral characteristics are primary drivers of soil carbon stock and Yasso20's predictive bias, is interesting for the whole modelling community.

**Author's response**

Thank you for the constructive feedback and suggestions. We have considered each of your remarks and modified the manuscript accordingly. You can find below our responses to all your comments. Here, we refer to the line number of the revised manuscript with track changes.

The main weakness of the manuscript comes to the lack of details about the forest context, litter and soil carbon estimation and modelling procedures:

More detailed information on forest management (LN87) and disturbance history (LN88) would be appreciated. For example, how were tree branches harvested? (Models usually ask for this kind of size thresholds that can play a role in decomposition rates). Western Europe, including Switzerland, is known to have suffered from storms in the past, including the Lothar in 1999 (see https://www.wsl.ch/en/news/25-years-after-lothar-how-the-windstorm-rebuilt-the-forest/). So none of these forest plots were attacked?

**Author's response**

Thank you for your remarks. Our study focuses on forests older than 120 years (old-growth forests, identified by the use of historic maps; Gosheva et al., 2017). As the soil sampling was performed on undisturbed sites, none of the sampled 556 sites was impacted by the storms Vivian (1990) or Lothar (1999). Although most of the Swiss forests are regularly exploited (62% of the forests, up 79% in the Swiss Plateau), the harvesting practices normally consist of planned loggings of single trees with harvesting residues (typically the smaller branches and tree top, foliage and belowground part with trunk to approx. 30 cm aboveground) normally left in the stand (Brändli et al., 2020). This keeps the disturbance by management interventions at a minimum. We are aware that natural disturbances from windthrows (i.e. Vivian and Lothar) in Swiss forests led to SOC losses especially in the organic layers of forests at high elevation (1 to 2 kg C m$^{-2}$) (Mayer et al., 2023), but none of the sampled sites were impacted.

**We have now added this information at lines 89-95 in the revised manuscript with track changes:**

"*In combination with the typically small-scale forest management in Switzerland, which includes planned loggings of single trees with harvesting residues (e.g. fine branches, foliage, belowground part with trunk to approx. 30 cm aboveground) normally left in the stand (Brändli et al., 2020), the focus on old forests ensures that there have been only reduced disturbances in the forest cover over the past decades. Although natural disturbances such as windthrows occurred in Swiss forests (e.g. Vivian in 1990 and Lothar in 1999) with possible impact on SOC stocks (Mayer et al., 2023), none of the forest sites was affected as sampling excluded windthrow sites.*"

More detailed information should be provided on how the NPP (LN150-1) was obtained (please avoid only citing other papers) and, especially, how it was converted into the estimate of annual litter input. Since litter input is derived from model outputs, so what is the suitability and prediction quality of the model (Terra and Aqua MODIS-satellite) for Swiss forests? This is not mentioned in LN152-153. As a result, one might ask about the reliability of the input, especially when reading LN410-411.

**Author's response**

Thank you for your comment. Since no stand inventory or litter input measurements were available at the investigated soil sites, we obtained the NPP from Terra and Aqua MODIS satellite (500-m resolution). We considered the NPP as a proxy of long-term litter C input to the soil, which was used to simulate SOC stocks at steady state with Yasso (see line 159-166). This approach is consistent with the published SOC model applications (Abramoff et al., 2022; Pierson et al., 2022), and with the current Yasso20 calibration (Viskari et al., 2022), where the litter input was similarly obtained from the GPP product of Terra and Aqua MODIS satellite, setting a maximum NPP to GPP ratio to 0.5. In our study, we also considered a maximum NPP to GPP ratio of 0.5, which agrees well with the NPP to GPP ratio (0.42-0.49) observed at two contrasting forest ecosystems in Switzerland (Etzold et al., 2011).

We also tested the suitability of Terra and Aqua MODIS-satellite to estimate NPP at Swiss forests by comparing the NPP obtained by Terra and Aqua MODIS-satellite with NPP derived by stand inventory data at 18 Swiss forest sites of the long-term forest ecosystem research LWF (Etzold et al., 2014). On average for the period 2001-2010, the mean NPP estimated by the satellite approach was $0.49\pm0.04$ kg C m$^{-2}$ yr$^{-1}$, while a mean NPP of $0.46\pm0.05$ kg C m$^{-2}$ yr$^{-1}$ was estimated by the terrestrial approach for the same period (Etzold et al., 2014). In addition, the satellite NPP proved to be a reasonable proxy of large range of forest productivity across Swiss forests (see new **Fig. S1a**, which has been added to the supplement), consistently with the wood increment trends across Swiss forest regions shown by the Swiss NFI (Brändli et al., 2020).

Taken together, these results confirm the suitability of the NPP obtained Terra and Aqua MODIS-satellite as a proxy of long-term litter inputs for Swiss forest conditions.

**We have now expanded and clarified in the manuscript the limitations of the approaches (satellite and terrestrial) to estimate litter inputs at lines 437-457 in the discussion of the revised manuscript, and added Fig. S1a to the supplement:**

*"Since measurements of forest stands and soil C inputs are often lacking at larger scales - as in this study - the satellite-derived NPP is here used as proxy of long-term litter C input to the soil, consistently with SOC model applications at the regional and global scales (Abramoff et al., 2022; Pierson et al., 2022), as well as with the calibration of Yasso20 (Viskari et al., 2022). Uncertainty in litter inputs potentially contributed to the observed discrepancies between simulated and measured SOC stocks at the site level. The fine scale variability in litter inputs cannot be captured by satellite-derived NPP estimates given (1) the larger pixel size of MODIS (500 m x 500 m) compared to the site scale of the soil sampling, and (2) the partitioning into tree components using average allocation factors, due to the lack of site-level data. NPP estimates from MODIS may overestimate the litter input in regions with intensive forest management as in the Plateau, since small-scale disturbances such as thinning are not well detected by satellites (Neumann et al., 2015; Park et al., 2021). Lastly, forests allocate a portion of NPP not only to fast-cycling components that are annually returned to the soil (i.e. fine roots and foliage) but also to components with slower turnover time such as stems and branches. Nevertheless, the satellite approach proves to be a reasonable proxy of the large range of forest productivity across Swiss forests, i.e. ranging from 0.3 kg C m$^{-2}$ yr$^{-1}$ in the Alps to 0.8 kg C m$^{-2}$ yr$^{-1}$ at the warmest sites (see Fig. S1a), which is consistent with differences in wood increments across regions as shown in the Swiss NFI (Brändli et al., 2020). Moreover, at the 18 sites of the long-term forest monitoring program LWF, the mean NPP over the period 2001-2010 estimated by MODIS satellite amounted to $0.49\pm0.04$ kg C m$^{-2}$ yr$^{-1}$ as compared to $0.46\pm0.05$ kg C m$^{-2}$ yr$^{-1}$ estimated by a terrestrial approach for the same period (Etzold et al., 2014). Terrestrial methods based on forest inventories may also produce uncertain estimates of litter inputs. These uncertainties mostly relate to (1) country-specific allometries and expansion factors used to estimate tree biomass, (2) turnover times applied to derive the annual litter inputs, and (3) failing to appropriately estimate inputs from fine roots and understory vegetation, which remain*

*severely unconstrained despite their major contribution to forest soil C inputs (Didion, 2020; Neumann et al., 2020)."*

[Figure]

**Fig. S1a**. *Net primary production (NPP) across Swiss forest regions, excluding waterlogged soils. Total n sites = 468. Letters indicate significantly different across regions, based on ANOVA followed by Tukey's test with P < 0.05.*

Then, how was tree litter chemical quality split? It is not very clear in the text (LN158-160). Was it partitioned at the level of tree species or at the level of tree functional type (broadleaved versus coniferous)?

Also, so a fixed set of allocation factors were used to estimate tree component (LN155) without distinguishing between tree functional type?

**Author's response**

We appreciate your remarks and suggestions. The litter chemical quality was split into AWEN fractions based on the percentage of broadleaf and conifer species at each site recorded by field assessments (as described at lines 123-125). First "*we partitioned the NPP into broadleaf and conifer species, multiplying the NPP by the percentage of broadleaf and conifer species recorded by field assessments at each site*" (see lines 161-163). Then the NPP – after being partitioned into broadleaf-derived and conifer-derived NPP - was split into AWEN fractions using the functions and approach used in the Swiss GHG inventory (Didion, 2023) based on measured fractions in the long-term forest monitoring program LWF (Didion et al., 2014).

**We have now clarified the methodology at lines 167-170 in the revised manuscript:**

*"The C inputs for each pool were separated into the four AWEN components according to the percentage of broadleaf and conifer species at each site, using the functions and approach adopted in the Swiss GHG inventory (Didion, 2023) based on measured fractions at sites of the long-term forest monitoring program LWF (Didion et al., 2014)."*

In this study, the estimates of NPP allocation into tree components rely on long-term forest ecosystem data from LWF network (Etzold et al., 2014), see lines 163-166. We did not distinguish between tree species or tree functional types, since we did not observe major differences between different tree functional types in C allocation to different tree compartments. In the future, this approach can be further refined by establishing species-specific C allocation factors, which should also consider the effect of climatic conditions on C allocation.

I find the categorization (LN119) risky. In this case, stands of 40-49% broadleaf and stands of 51-60% broadleaf which are actually quite similar stands in term of species mixture, will belong to two contrasting categories. I wonder if a new category of "mixed stand" should be proposed and compared with more pure stands? Also, how was the chemical quality of the litter in these mixed stands partitioned? As a reader, I am also curious to know how big a difference that the effect of species functional type can theoretically make in Yasso20 simulations in your case? (According to Fig. 3b, there is limited difference in the model's predicts). Would it be possible to do an uncertainty analysis on how the 0%, 50% and 100% of broadleaves at a given site (with given climatic inputs) can change the predictions? This can be done for all sites or only for those where both broadleaves and conifers coexist, enabling us to better understand the sensitivity of the species related parameters used in your case to the model predictions.

**Author's response**

Thank you for your suggestions. In the manuscript, we have separated forest sites into broadleaf and coniferous forests only for visualization purposes in the Figures (e.g. Fig. 2-4). The percentage of broadleaf species recorded at each site has already been considered for partitioning NPP input at each site, which is then used as model input for Yasso20 simulations. We have specified this at lines 161-163 of the manuscript: *"We partitioned the NPP into broadleaf and conifer species, multiplying the NPP by the percentage of broadleaf and conifer species recorded by field assessments at each site"*.

**To improve clarity, we have now rephrased the sentence at lines 126-127 in the revised manuscript:**

*"Only for visualization purposes, the forest sites were subdivided into two types based on the broadleaf percentage: coniferous (0-50%) and broadleaf (51-100%) forests."*

Finally, in terms of data representation and visualization, the authors chose to present simulated versus observed C using bar plots (Fig. 3), box plots (Fig. S1) or scatter plots (deviation versus factors as in Fig. S3). While it is good, it would be more informative to directly show the scatter plots of simulated C versus observed C at the site level with a 1:1 line to see if and how the model captures the observations according to different tree functional types (and/or regions). A new (or supplementary) plot at the site level would be desirable as a complement to Fig. 3.

**Author's response**

We presented the simulated vs observed C stocks using bar plots to better visualize in which regions the simulated SOC stocks deviate more from the measured stocks, in order to infer on the main processes driving mineral stabilization.

**We have now added the 1:1 plot of simulated vs observed C stocks in the supplemental material (see Fig. S2 below, which has been added to the supplement)**. As expected, the Yasso20-simulated values show only a smaller variability in SOC stocks compared to the measured SOC across sites, although the average simulated SOC stocks match well the average SOC stocks across Swiss forest soils. Possible reasons for this are: (1) the uncertain litter input estimation, with the use of satellite-derived NPP as a proxy of site productivity (see paragraph added in the revised manuscript at lines 437-457), (2) the high spatial variability of soil properties which drives very high SOC stocks at some sites (e.g. high Fe and Al), and (3) the anaerobic conditions at the site scale that retard the SOC mineralization rates and lead to locally higher SOC stocks.

[Figure]

*Fig. S2.* *Comparison between Yasso20-simulated and measured SOC stocks by forest types, with 1:1 dotted line. The measured SOC stocks are the sum of SOC stocks in organic layers and mineral soils to 100 cm depth (total n sites = 468, excluding waterlogged soils). The simulated SOC stocks at each site are based on the mean of 500 replicate simulations representing model parameters uncertainty.*

Specific points:

LN88: "minimal disturbances" was mentioned, but latter we read "frequent fires" (LN354). Fires are an important disturbance in the forest system. Is this a contradiction? What about storms (see above)?

**Author's response**

Thank you for the comment. We have included in the manuscript your remark that disturbances such as forest management or windthrows may affect soil C in Swiss forests at line 93-95 (*see previous Author's response*), although none of the sampled site was directly affected by windthrows. In Switzerland, forest fires - including wildfires during the dry season and deliberate fires common especially in the Southern Alps for over 10,000 years – contributed to increase SOC storage due to increased resistance of charred material to decomposition (Eckmeier et al., 2010). High amounts of charred material persisted in Fe- and Al-rich soils for millennia (Eckmeier et al., 2010). Here, we cannot exclude that the sampled sites have not been affected by forest fires over millennia time scales. Thus, we already discussed in the manuscript at line 372-377 how fire-derived black C may potentially affect soil C stock measured in the Southern Alps, showing that the main results remained unchanged even by excluding this region from the analysis.

LN125: how were the percentages of C content of the litter measured?

**Author's response**

The C content of ground organic materials (including the following horizons: undecomposed litter L, fermentation F, and humified H) and mineral soil were measured by dry combustion using an elemental analyzer (NC 2500, CE Instruments, Italy). This is reported at line 113-114.

**To improve clarity, we have added the following sentence in the revised manuscript, see lines 133 and 137:**

"*obtained by dry combustion (see section 2.2).*"

LN129: it would be good to explain how the volumetric stone content was estimated.

**Author's response**

Thank you for the comment. **We have added this information in the manuscript at line 138 in the revised manuscript:**

"*volumetric stone content ($m^3 \ m^{-3}$) visually estimated from the soil profile (Richard et al., 1978)*"

LN183: also with a standard deviation of 1 ?

**Author's response**

We have centered all independent variables before analysis to a mean of 0. We did not scale the variables to standard deviation of 1. This allows to provide in Table 2 and 3 (column Est.) meaningful model estimates, which can be interpreted in the same measurement units of the predictor and response variables.

**To improve clarity, we have replaced the word "standardized" with "centered" in the sentence at line 195 in the revised manuscript:**

"*All explanatory variables were centered to a mean of 0*"

LN 320: very interesting result for clay. Have you also tried the sand content (which does not necessarily correlate with the clay content)?

**Author's response**

We tested whether the sand content would be significantly correlated to the SOC stocks, which we plot here below for your information. No significant correlation was found between sand content and SOC stocks (where r is the Person correlation coefficient, $r = -0.04$, $p = 0.20$, $n = 468$ sites), similarly to the clay content:

[Figure]

In addition, a strong correlation of sand and clay content was observed in this study ($r = -0.83$, $p < 0.001$, $n = 468$ sites):

[Figure]

For these reasons, we have decided to include only the clay content (%) in the analysis presented in the manuscript.

Table 2 (and also other Tables that summarize statistics), I suggest using a MAP unit of 100 mm rather than mm to give a more meaningful sense for each increment of 1 unit.

**Author's response**

Thank you for your suggestion. **We have modified the Tables summarizing statistical analysis using a MAP unit of 100 mm instead of 1 mm, see updated Table 2, Table 3, Table S2, Table S3, Table S4, Table S5, Table S6.**

We also often see the important factor of "slope", but it is rarely mentioned. And why is that?

**Author's response**

We agree that the slope can be an important driver of SOC stocks. In fact, we have included slope as a predictor in our statistical analysis. In the manuscript at lines 291-293, we report that slope has a negative effect on SOC stocks. To not further increase the length of our manuscript and considering that slope did not appear a dominant driver of SOC stocks in the different regions (Table S5), we do not discuss about slope in more detail. However, topography represents an important factor influencing SOC stocks, with soil erosion occurring on steep slopes while waterlogging in flat areas (Fernández-Romero et al., 2014) which can strongly impact the variability of SOC stocks at the site-scale.

Figure 2: Why was a polynomial regression also used? Would it be possible to plot the litter input data in addition to NPP?

**Author's response**

We have shown a polynomial regression in Fig. 2a in addition to the linear correlation, since it allows to better capture the productivity trend across elevation, with NPP peaking at an elevation of about 700 m a.s.l., and then decreasing with increasing elevation (line 224-225).

Since we do not have the litter input at the sites, we considered the NPP as a proxy of long-term litter carbon input to the soil, see previous *Author's response* and revised manuscript at lines 437-457. We have now plotted the NPP data in the new **Fig. S1a added to the supplement**, which show the variability in NPP across Swiss forest regions.

Figure 3: I have the impression that the Yasso20 estimates for all five regions are not very different and possibly not very significant in most regions. In (b) this may also be the case: does Yasso20 predict significantly different stocks between broadleaved and coniferous stands? It would also be interesting to read statistical diagnoses somewhere among the simulated (or observed) groups.

**Author's response**

In the **updated Fig. 3a,b** (see below and in the revised manuscript), we have now added results of statistical analysis with letters that indicate significantly different means (1) among measured SOC stocks (capital letters), and (2) among Yasso20-simulated SOC stocks (lowercase letters).

We show that Yasso20-simulated SOC stocks are statistically different across regions (**Fig. 3a**) and between forest types (see **Fig. 3b**), but differences are more limited compared to differences observed in measured SOC stocks. Yasso20-simulated SOC stocks across regions show a different pattern compared to observed SOC stocks, with Jura and Pre-Alps > Southern Alps. We regard the smaller variation of SOC stocks modelled by Yasso20 as an indication for the lacking representation of region-specific soil characteristics such as mineralogy.

**We have modified Fig. 3a,b and its caption, which now includes also the comparison among measured (capital letters), and among simulated SOC stocks (lower case letters):**

[Figure]

**Figure 3:** Comparison between measured SOC stocks vs Yasso20-simulated stocks by: (a) regions of Switzerland, with waterlogged soils shown separately, and (b) forest types, excluding waterlogged soils. The measured stocks are shown as organic layers and mineral soils to 100 cm depth (total $n$ sites = 556; excluding waterlogged soils, $n$ sites = 468). The simulated stocks at each site are based on the mean of 500 replicate simulations representing model parameters uncertainty. SOC stocks are represented as means ± standard errors. $P$-values are calculated with Welch's $t$-tests: $P \geq 0.05$ (n.s., not significant), $P <$ 0.001 (***). **Capital letters indicate significantly different means among measured SOC stocks, while lowercase letters indicate significantly different means among Yasso20-simulated SOC stocks, based on ANOVA followed by Tukey's test (Fig. 3a) and Welch's t-tests (Fig. 3b) with $P$ < 0.05.**

Fig. 4: It would be good to briefly explain why such a subjective cut-off point of pH = 5 was chosen for data analysis. It is particularly useful for non-soil experts, although it is intuitively logical for soil experts.

**Author's response**

Thank you for the remark. **This clarification was added at lines 272-274 in the revised manuscript:**

"*We separated the dataset into soils with pH ≤ 5 and with pH > 5, since pH is recognized as a predictor of SOC stabilization mechanisms, mediated by Al- or Fe- under acidic conditions while Ca exerts a dominant influence under increasing pH (Rowley et al., 2018).*"

**Anonymous Referee #2, 15 Feb 2025**

Review of" Drivers of soil organic carbon from temperate to alpine forests: a model-based analysis of the Swiss forest soil inventory with Yasso20".

Authors analyzed significance of soil properties especially missing organo-mineral (OM) interactions on mismatch between the soil organic C (SOC) stocks measured in Alps and their modeled counterparts estimated with Yasso20 model. The study itself is interesting and relevant.

The conclusion is in line with the expectations of the theoretical set up of the analysis. Authors conclude that the model failed to estimate SOCs in regions where OM is relevant mechanism of total SOC accumulation and where dependency of decomposition on precipitation could not capture its reduction related to higher soil moisture levels. Authors thus demonstrate that the model indeed fails to estimate precisely the SOC variability and conclude that if the model is applied in Alps, then it needs further development. This is known in literature and not new.

What is new is nicely demonstrating where the model **(applied as in this study)** fails in Alps and how much it fails. Simplified, these are the soils with ph <5 showing Fe correlation to SOC and ph > 5 showing Ca correlation to SOC, and soils with MAP > 1400 mm.

**Author's response**

Thank you for the constructive feedback and suggestions. We have considered each of your remarks and modified the manuscript accordingly. You can find below our responses to all your comments. Here, we refer to the line number of the revised manuscript with track changes.

However, authors also demonstrate that the SOC discrepancies between measurements and Yasso20 could be estimated more precisely with statistical models (explaining about 50% of variance). Although, how much of total variance is explained in combination of Yasso + statistical models is not known. The final conclusions revolve about findings expected from the analysis setup. The motivation why SOC needs to be evaluated in combination of statistical and process models is not clear.

**Author's response**

Thank you very much for your suggestion. In this study, we make use of the deviations between Yasso-simulated SOC stocks and measured SOC stocks to more clearly disentangle the main drivers of SOC stocks across an extensive Swiss forest soils dataset. Since Yasso is a simple soil C model driven by litter inputs and climate, deviations between simulated and measured SOC stocks should be mostly related to drivers of SOC stocks not accounted in the models, such as mineral soil properties, which are not explicitly implemented in Yasso. For this reason, we have sequentially analyzed the Yasso deviations by statistical model, which allows us to evaluate the effects of different predictors on Yasso deviations.

Although a combined approach – directly coupling Yasso with a statistical model - would be very promising since it will allow to incorporate predictors that are not explicitly included in the Yasso model (e.g. mineral soil properties), this would be beyond the focus of this manuscript. As a first step in this direction, we here demonstrate what are the: (1) main drivers of SOC stocks, and (2) main drivers of Yasso20 deviations for the Swiss forest soils. This could contribute to further improve Yasso in future works.

**We have now clarified the rationale for using Yasso followed by statistical analysis at lines 69-75 in the introduction of the revised manuscript**:

*"Here, we aimed to identify the main factors controlling SOC stocks in Swiss forest soils across a large gradient of climate, soil biogeochemistry and forest types. To disentangle the main drivers of SOC stocks, our main approach was to (1) simulate SOC stocks in forest soils by Yasso20, driven by litter input and climate, and (2) statistically analyze the deviations between Yasso20-simulated and measured SOC stocks. This allows to evaluate the importance of mineral-driven SOC stabilization, since mineral soil properties are not explicitly implemented in Yasso."*

**We have added the suggestion of a combined approach beween Yasso and a statistical model at lines 469-471 in the revised discussion:**

"*A combined approach – directly coupling Yasso with a statistical model - would allow to account for additional parameters (as mineral soil properties) that are currently not included as model drivers but are known to be important factors controlling SOC stabilization.*"

All mismatch between Yasso and measurements of SOC is blamed on Yasso missing relevant processes in the model structure of decomposition, though the litter C input itself which is the most essential driver of Yasso modeled SOC was estimated using rather simple imprecise approach with high level of uncertainty. **Given that the C inputs were estimated with low precision (litter inputs C quantity represented average for all species, litter input C quality differing only between conifers and broadleaves, and C input representing rather large spatial unit of 0.25 km2 derived from last 20 years compared to soil samples representing much smaller unknown unit cc 10 m2 representing SOC development changes of 1000+ years?) this should be taken into consideration when interpreting the modeled SOC data. At present state, the study is biased by ignoring impact of imprecise estimates of C input data used to run Yasso.**

**Author's response**

Thank you very much for your remark. We are aware that NPP represent an approximation of the productivity at each site and a proxy of long-term C input to the soil. Unfortunately, no stand inventory or litter input measurements were available at the sampled sites.

Here, we considered the NPP (from Terra and Aqua MODIS satellite at 500-m resolution) as a proxy of long-term litter C input to the soil, which was used to simulate SOC stocks at steady

state with Yasso (see line 159-163). This is consistent with the approach of published SOC model applications (Abramoff et al., 2022; Pierson et al., 2022), and with the current Yasso20 calibration (Viskari et al., 2022), where NPP is used as model input.

Considering that NPP represents an approximation of long-term litter C input to the soil, we address the limitations of this satellite approach in the updated manuscript. In addition, we discuss that terrestrial-based methods may also lead to biased estimates of litter inputs, e.g. due to the application of country-specific allometric equations or due to underestimates of C inputs from rhizosphere or understory vegetation. Both satellite- and terrestrial-based approaches to estimate litter inputs are normally obtained from the last few decades of measurements, while SOC turns over on decadal to millennia timescales.

Despite the difficulties with any approach in estimating C inputs, we are certain that the large-scale pattern in C inputs across Swiss forests with MATs ranging between 2 and 12°C and MAPs ranging between 400 and 2200 mm is robust. Therefore, also the magnitude of the differences between regions explored here appears robust.

**We have now expanded and clarified in the manuscript the limitations of the approaches (satellite and terrestrial) to estimate litter inputs at lines 437-457 in the revised manuscript, and added Fig. S1a to the supplement, showing the range in NPP across Swiss forest regions:**

*"Since measurements of forest stands and soil C inputs are often lacking at larger scales - as in this study - the satellite-derived NPP is here used as proxy of long-term litter C input to the soil, consistently with SOC model applications at the regional and global scales (Abramoff et al., 2022; Pierson et al., 2022), as well as with the calibration of Yasso20 (Viskari et al., 2022). Uncertainty in litter inputs potentially contributed to the observed discrepancies between simulated and measured SOC stocks at the site level. The fine scale variability in litter inputs cannot be captured by satellite-derived NPP estimates given (1) the larger pixel size of MODIS (500 m x 500 m) compared to the site scale of the soil sampling, and (2) the partitioning into tree components using average allocation factors, due to the lack of site-level data. NPP estimates from MODIS may overestimate the litter input in regions with intensive forest management as in the Plateau, since small-scale disturbances such as thinning are not well detected by satellites (Neumann et al., 2015; Park et al., 2021). Lastly, forests allocate a portion of NPP not only to fast-cycling components that are annually returned to the soil (i.e. fine roots and foliage) but also to components with slower turnover time such as stems and branches. Nevertheless, the satellite approach proves to be a reasonable proxy of the large range of forest productivity across Swiss forests, i.e. ranging from 0.3 kg C m$^{-2}$ yr$^{-1}$ in the Alps to 0.8 kg C m$^{-2}$ yr$^{-1}$ at the warmest sites (see Fig. S1a), which is consistent with differences in wood increments across regions as shown in the Swiss NFI (Brändli et al., 2020). Moreover, at the 18 sites of the long-term forest monitoring program LWF, the mean NPP over the period 2001-2010 estimated by MODIS satellite amounted to 0.49±0.04 kg C m$^{-2}$ yr$^{-1}$ as compared to 0.46±0.05 kg C m$^{-2}$ yr$^{-1}$ estimated by a terrestrial approach for the same period (Etzold et al., 2014). Terrestrial methods based on forest inventories may also produce uncertain estimates of litter inputs. These uncertainties*

*mostly relate to (1) country-specific allometries and expansion factors used to estimate tree biomass, (2) turnover times applied to derive the annual litter inputs, and (3) failing to appropriately estimate inputs from fine roots and understory vegetation, which remain severely unconstrained despite their major contribution to forest soil C inputs (Didion, 2020; Neumann et al., 2020)."*

[Figure]

*__Fig. S1a__. Net primary production (NPP) across Swiss forest regions, excluding waterlogged soils. Total n sites = 468. Letters indicate significantly different across regions, based on ANOVA followed by Tukey's test with P < 0.05.*

From the study it is difficult to understand whether it makes sense at all to use Yasso model. Although, if Yasso's decomposing of fraction of NPP explains anything than it is somewhat hinted that the model combination would yield better results than statistical models only. The main strength of combination of Yasso and statistical models over only statistical models is, however, the potential for modelling not only SOC stocks but also their changes over time (needed for estimating of ecosystem C fluxes in warming climates).

**Author's response**

To evaluate the effects of the main drivers of SOC stocks, our approach was to analyze Yasso deviations from measured SOC stocks using a statistical model. An improvement of Yasso20 model may represent the next step. Despite the regional differences, Yasso20 captured the average SOC stocks for Switzerland. Please also refer to our *Author's response* above. As reported in the manuscript at lines 477-478, we do not simulate SOC stock changes here, since no repeated SOC stock measurements were available at the sites.

**We have now clarified the rationale for using Yasso followed by statistical analysis at lines 69-75 in the introduction of the revised manuscript** (see also response above). **We also added a clarification at lines 188-189 in the revised manuscript to facilitate result interpretation:**

*"Statistical analysis of Yasso20 deviations allows to assess effects of the main drivers of SOC stocks on model discrepancies from measured values".*

In theory in ideal application of Yasso, the litter input C quantity and quality should reflect implicitly the site environmental conditions including the soil properties. However, from the analysis here it could be expected that litter inputs represented only the average for the region and thus the SOC estimates also only represent the average and could not be expected to show the small-scale variability. For example, it can be expected that if you plot derived C inputs on 2$^{nd}$ axis in Fig.3 these would be only similar averages for different regions with not much variation as modeled SOCs, unlike variation seen in measured SOCs.

**Author's response**

While we agree with the reviewer that Yasso ideally captures litter inputs driven by environmental variables, it cannot capture all effects of soil properties on SOC stocks that lead to SOC stabilization in the soils (mineralogy) or suppress decomposition (anaerobic conditions) as they are not explicitly included in the model.

We are also aware that the fine scale variability in litter input cannot be fully detected by satellite-derived NPP estimates given the larger pixel size of MODIS (500 m x 500 m) (see *Author's response* above, and the changes in the manuscript already reported above, see lines **437-457**). However, the satellite NPP approach still provides a proxy of forest productivity across Swiss forests, being able to show a large productivity range across Swiss regions (see added **Fig. S1a**), which is consistent with the productivity trend across Swiss forest regions (Brändli et al., 2020). Given the low temperatures occurring in the least productive region (i.e. the Alps) – leading to slower SOC decomposition – and the high temperature at the most productive regions – leading to faster SOC decomposition – differences in simulated SOC stocks across regions were rather limited (see added **Fig. S1b**).

**We have added a supplemental figure which show the differences in NPP across regions, see new Fig. S1a, b added to the supplement:**

[Figure]

**Fig. S1**. *Net primary production (NPP) (a), and Yasso20-simulated SOC stocks (b) across Swiss forest regions, excluding waterlogged soils. Total n sites = 468. Letters indicate significantly different across regions, based on ANOVA followed by Tukey's test with P < 0.05.*

However, the main message is correct, as it demonstrates that Yasso model structure needs development for application in Alp and thus should account for interaction with minerals and limitation of decomposition with excessive moisture. Discussion claims that Yasso20 also fails due to missing moisture limitation (which is known) but does not notice that dependency on moisture was evaluated with Yasso07 (Tupek et al., 2024) and could have been already applied here instead of precipitation.

**Author's response**

Thank you very much for your suggestion of this promising approach to include soil moisture as model driver in Yasso, which can be especially useful for peatland forest C modelling (Ťupek et al., 2024). Given that the moisture modifier of Yasso07 was developed specifically for nine sites in a boreal forest–mire ecotone in Finland (Ťupek et al., 2024), the applicability of this approach would require more tests in a larger number of experimental study sites covering Swiss forests, before being applied to this large Swiss forest soil dataset. This would go beyond the aim of this paper.

**We have taken your remark into account at lines 395-397 in the revised manuscript:**

"*... could be resolved in the future by including soil moisture at monthly time steps as model driver, or by applying a moisture modifier as in a boreal forest–mire ecotone in Finland for Yasso07 (Ťupek et al., 2024), or by coupling Yasso to a soil water model (Guenet et al., 2024).*"

My main concern is that the relation of modeled SOCs on poor litter input estimates should be elaborated in the paper to separate the mismatch in measured and modeled SOCs due to Yasso model structure (misrepresented decomposition rates) and due to misrepresented C inputs. Showing that NPP did not correlate with SOCs (Tabe 2, and Fig. 3) does not help with confidence in estimated C inputs (which were not shown). Although, I recognize enormous work done and appreciate that the data used in the analysis was made openly available, the lack of confidence in litter input is the main weakness that requires clarifying in major revision before the paper can be accepted.

**Author's response**

Thank you very much for your comment. We have now elaborated in the manuscript that the discrepancies in the model residuals (simulated – measured SOC stocks) can also be related to uncertain litter input estimates (see *Author's response above*), in addition to the lack of implementation of soil properties and waterlogging processes in the current model formulation. Currently, we cannot fully disentangle the two components of uncertainties e.g. from litter input estimates and lack of soil processes representation in the model, but we have considered the uncertainty derived from litter input estimation by adding an additional paragraph to the manuscript.

**We have now expanded and clarified in the revised manuscript the limitations of the approaches to estimate litter inputs at lines 437-457 in the discussion, and added Fig. S1a to the supplement, which shows the NPP range across Swiss regions:**

*"Since measurements of forest stands and soil C inputs are often lacking at larger scales - as in this study - the satellite-derived NPP is here used as proxy of long-term litter C input to the soil, consistently with SOC model applications at the regional and global scales (Abramoff et al., 2022; Pierson et al., 2022), as well as with the calibration of Yasso20 (Viskari et al., 2022). Uncertainty in litter inputs potentially contributed to the observed discrepancies between simulated and measured SOC stocks at the site level. The fine scale variability in litter inputs cannot be captured by satellite-derived NPP estimates given (1) the larger pixel size of MODIS (500 m x 500 m) compared to the site scale of the soil sampling, and (2) the partitioning into tree components using average allocation factors, due to the lack of site-level data. NPP estimates from MODIS may overestimate the litter input in regions with intensive forest management as in the Plateau, since small-scale disturbances such as thinning are not well detected by satellites (Neumann et al., 2015; Park et al., 2021). Lastly, forests allocate a portion of NPP not only to fast-cycling components that are annually returned to the soil (i.e. fine roots and foliage) but also to components with slower turnover time such as stems and branches. Nevertheless, the satellite approach proves to be a reasonable proxy of the large range of forest productivity across Swiss forests, i.e. ranging from 0.3 kg C $m^{-2}$ $yr^{-1}$ in the Alps to 0.8 kg C $m^{-2}$ $yr^{-1}$ at the warmest sites (see Fig. S1a), which is consistent with differences in wood increments across regions as shown in the Swiss NFI (Brändli et al., 2020). Moreover, at the 18 sites of the long-term forest monitoring program LWF, the mean NPP over the period 2001-2010 estimated by MODIS satellite amounted to 0.49±0.04 kg C $m^{-2}$ $yr^{-1}$ as compared to 0.46±0.05 kg C $m^{-2}$ $yr^{-1}$ estimated by a*

*terrestrial approach for the same period (Etzold et al., 2014). Terrestrial methods based on forest inventories may also produce uncertain estimates of litter inputs. These uncertainties mostly relate to (1) country-specific allometries and expansion factors used to estimate tree biomass, (2) turnover times applied to derive the annual litter inputs, and (3) failing to appropriately estimate inputs from fine roots and understory vegetation, which remain severely unconstrained despite their major contribution to forest soil C inputs (Didion, 2020; Neumann et al., 2020)."*

[Figure]

***Fig. S1a***. *Net primary production (NPP) across Swiss forest regions, excluding waterlogged soils. Total n sites = 468. Letters indicate significantly different across regions, based on ANOVA followed by Tukey's test with P < 0.05.*

References:

Ťupek, B., Lehtonen, A., Yurova, A., Abramoff, R., Guenet, B., Bruni, E., Launiainen, S., Peltoniemi, M., Hashimoto, S., Tian, X., Heikkinen, J., Minkkinen, K., and Mäkipää, R.: Modelling boreal forest's mineral soil and peat C dynamics with the Yasso07 model coupled with the Ricker moisture modifier, Geosci. Model Dev., 17, 5349–5367, https://doi.org/10.5194/gmd-17-5349-2024, 2024.

**References - Reply to Editor and Reviewers**

Abramoff, R.Z., Guenet, B., Zhang, H., Georgiou, K., Xu, X., Rossel, R.A.V., Yuan, W., Ciais, P., 2022. Improved global-scale predictions of soil carbon stocks with Millennial Version 2. Soil Biology and Biochemistry 164, 108466.

Brändli, U.-B., Abegg, M., Allgaier, B.L., 2020. Schweizerisches Landesforstinventar: Ergebnisse der vierten Erhebung 2009-2017. WSL.

Didion, M., 2020. Extending harmonized national forest inventory herb layer vegetation cover observations to derive comprehensive biomass estimates. Forest Ecosystems 7, 1-14.

Didion, M., 2023. Data on soil carbon stock change, carbon stock and stock change in surface litter and in coarse dead wood prepared for the Swiss NIR 2024 (GHGI 1990–2022).

Didion, M., Frey, B., Rogiers, N., Thürig, E., 2014. Validating tree litter decomposition in the Yasso07 carbon model. Ecological modelling 291, 58-68.

Eckmeier, E., Egli, M., Schmidt, M., Schlumpf, N., Nötzli, M., Minikus-Stary, N., Hagedorn, F., 2010. Preservation of fire-derived carbon compounds and sorptive stabilisation promote the accumulation of organic matter in black soils of the Southern Alps. Geoderma 159, 147-155.

Etzold, S., Ruehr, N.K., Zweifel, R., Dobbertin, M., Zingg, A., Pluess, P., Häsler, R., Eugster, W., Buchmann, N., 2011. The carbon balance of two contrasting mountain forest ecosystems in Switzerland: similar annual trends, but seasonal differences. Ecosystems 14, 1289-1309.

Etzold, S., Waldner, P., Thimonier, A., Schmitt, M., Dobbertin, M., 2014. Tree growth in Swiss forests between 1995 and 2010 in relation to climate and stand conditions: Recent disturbances matter. Forest Ecology and Management 311, 41-55.

Fernández-Romero, M., Lozano-García, B., Parras-Alcántara, L., 2014. Topography and land use change effects on the soil organic carbon stock of forest soils in Mediterranean natural areas. Agriculture, Ecosystems & Environment 195, 1-9.

Fischer, C., Traub, B., 2019. Swiss National Forest Inventory - Methods and Models of the Fourth assessment. Managing forest ecosystems, 35 431 p. Springer.

Guenet, B., Orliac, J., Cécillon, L., Torres, O., Sereni, L., Martin, P.A., Barré, P., Bopp, L., 2024. Spatial biases reduce the ability of Earth system models to simulate soil heterotrophic respiration fluxes. Biogeosciences 21, 657-669.

Mayer, M., Rusch, S., Didion, M., Baltensweiler, A., Walthert, L., Ranft, F., Rigling, A., Zimmermann, S., Hagedorn, F., 2023. Elevation dependent response of soil organic carbon stocks to forest windthrow. Science of the Total Environment 857, 159694.

Neumann, M., Godbold, D.L., Hirano, Y., Finér, L., 2020. Improving models of fine root carbon stocks and fluxes in European forests. Journal of Ecology 108, 496-514.

Neumann, M., Zhao, M., Kindermann, G., Hasenauer, H., 2015. Comparing MODIS net primary production estimates with terrestrial national forest inventory data in Austria. Remote Sensing 7, 3878-3906.

Park, J.H., Gan, J., Park, C., 2021. Discrepancies between global forest net primary productivity estimates derived from MODIS and forest inventory data and underlying factors. Remote Sensing 13, 1441.

Pierson, D., Lohse, K.A., Wieder, W.R., Patton, N.R., Facer, J., de Graaff, M.-A., Georgiou, K., Seyfried, M.S., Flerchinger, G., Will, R., 2022. Optimizing process-based models to predict current and future soil organic carbon stocks at high-resolution. Scientific Reports 12, 10824.

Richard, F., Lüscher, P., Strobel, T., 1978. Physikalische Eigenschaften von Böden der Schweiz. Eidgenössischen Anstalt für das forstliche Versuchswesen.

Rowley, M.C., Grand, S., Verrecchia, É.P., 2018. Calcium-mediated stabilisation of soil organic carbon. Biogeochemistry 137, 27-49.

Ťupek, B., Lehtonen, A., Yurova, A., Abramoff, R., Guenet, B., Bruni, E., Launiainen, S., Peltoniemi, M., Hashimoto, S., Tian, X., 2024. Modelling boreal forest's mineral soil and peat C dynamics with the Yasso07 model coupled with the Ricker moisture modifier. Geoscientific Model Development 17, 5349-5367.

Viskari, T., Pusa, J., Fer, I., Repo, A., Vira, J., Liski, J., 2022. Calibrating the soil organic carbon model Yasso20 with multiple datasets. Geoscientific Model Development 15, 1735-1752.

---

## Author Response (AR2)

**Author's response - "*Drivers of soil organic carbon from temperate to alpine forests: a model-based analysis of the Swiss forest soil inventory with Yasso20*"**

**Editor's comments**

Dear authors,

Both reviewers agree that your manuscript has been greatly improved. However, one of the reviewers still has some minor comments. If you address them, the manuscript will be accepted.

Best regards

Bertrand Guenet

**Author's response**

We thank the Editor and the Referees for their positive evaluation of the revised manuscript, and we appreciate the additional comments that you provided. We have addressed each of the remaining comments. The manuscript has been revised accordingly (see *Manuscript with track changes*).

You can find below our responses to the reviewer's comments, referring to the line number of the revised manuscript with track changes.

**Anonymous referee #1: accepted subject to minor revisions**

The authors have done a tremendous amount of work revising the manuscript. They have answered each of the points I raised in detail. I greatly appreciate the effort they have made. While I am satisfied with many of the responses, I still have some concerns.

The result in Fig. S2 (i.e. the 1:1 plot of simulated versus observed carbon stocks) is striking. Such an important result should be presented as a figure in the main text, rather than as a supplementary one (that most readers will not read), to truly demonstrate the model's performance. It appears that the predicted and measured data are not even correlated. This is fairly embarrassing for the model, suggesting that the comparable mean values achieved (LN22–23) may be coincidental.

**Author's response**

Thank you very much for the positive feedback on the revised manuscript, and for the further comments that you provided. Although the average simulated SOC stocks match well the average SOC stocks across Swiss forest soils (see Fig. 3), we are aware that the Yasso model and its application in this study does not capture the variability of SOC stocks at the site level (Fig. S2). This was expected, since the model does not consider: (1) mineral soil properties

driving SOC stabilization (see lines 72-73, 79-83 etc. in the manuscript), with minerals (e.g. Fe and Al) being highly variable at the site scale, and (2) anaerobic conditions that may retard SOC decomposition and lead to locally high SOC stocks (see lines 386-390 in the manuscript). Additionally, uncertainties in litter inputs (i.e. satellite-derived NPP) at the soil profile scale also potentially contributed to the observed discrepancies between simulated and measured SOC stocks at the site level (lines 430-431).

Here, we would prefer to keep the 1:1 plot of simulated vs observed C stocks in the supplemental material (Fig. S2) given that the main goal of the manuscript is to identify the main factors controlling SOC stocks in Swiss forests - and thus which processes should be included in soil C models - rather than to evaluate and demonstrate the model performance at the site level. We have added reference to Fig. S2 at lines 431 and 449 in the main text.

**Reviewer's comment**

My suggestions are as follows:

The conclusion (LN489–491) should be toned down considerably by emphasising that the model fails to capture variations at the site level. This is because the simulated and observed data are not only far from the 1:1 line, but are also poorly (or not at all) correlated.

**Author's response**

Thank you for your remark. We have now added the following sentence at line 481 in the conclusion: "*although the model does not capture the variability in SOC stocks at the site level*".

**Reviewer's comment**

I am also wondering whether a positive correlation between simulated and observed C stocks could be found at the regional level. While I suggest making Fig. S2 as the new Fig. 4, could it be possible to create a 1:1 scatter plot for each region as a supplementary figure to complement Fig. 3a too?

**Author's response**

Thank you for your suggestion. The correlation between simulated and observed SOC stocks is weak but significant across all sites (where r is the Person correlation coefficient, r = +0.08, $p$-value = 0.034, $n$ = 468 sites). Similarly, at the regional level, only weak or no correlation between simulated and measured SOC stocks was observed with the exception of the Southern Alps, likely due to the larger SOC stocks gradient found in this region (r = + 0.44, $p$-value = 0.006). We have not added this plot to the main manuscript or supplemental material, given that it is not adding more information compared to the Fig. S2 already presented. Moreover, we want to stress again that the main goal of the manuscript is to identify the main factors controlling SOC stocks and thus which processes should be accounted in soil C models rather than to evaluate the model performance at the site level.

**Reviewer's comment**

LN195: Ok, but if variables are only centred, but not standardised (with z-score normalisation), they may not be homoscedastic and therefore may violate the rule for certain analyses (such as Pearson's correlation, see Fig. 4).

**Author's response**

Thank you for the remark. We note that while centering and standardization serve specific purposes - such as reducing multicollinearity or making coefficients comparable - they do not directly ensure homoscedasticity. This assumption was instead verified visually through residual and scatter plots (line 192-193).

The centering of explanatory variables was performed only in linear mixed-effect models, not in Pearson's correlation analysis. Scaling the variables to a certain standard deviation (as done with standardization), would mean that the estimated coefficient changes by the corresponding factor. We choose to not scale the variables to a certain standard deviation since we wanted to provide in Table 2 and 3 (column Est.) meaningful model estimates, which can be interpreted in the same measurement units of the predictor and response variables and can be more easily taken into account in further studies.

**Reviewer's comment**

LN280 and LN192-4: Why do you sometimes transform the x-label data but not at other times? How did you decide on the transformation method?

**Author's response**

Thank you for your comment. Transformations such as log or square root were applied when the relationship between the predictor and response was non-linear or when residuals showed signs of heteroscedasticity or non-normality. The decision to transform variables was guided by visual inspection of scatterplots and residual plots (line 192-193).

To clarify how we decided on the transformation method, we have rephrased the text at line 189-192: "*The numerical explanatory variables were log- or square-root transformed when the relationship between the explanatory and the response variable was non-linear or when the residuals showed signs of heteroscedasticity or non-normality.*"

In Fig. 4 and Fig. S5, we show only for visualization purposes the exchangeable Fe and Al on a square-root scale x-axis, and clay on a natural-logarithm scale. Transformations applied in the linear mixed-effect models are reported in the Tables where results of statistical analysis are shown (see Table 2, Table 3, Table S2-S6).

**Anonymous referee #2: accepted as is**

Authors sufficiently addressed my main concern's about the uncertainty of derived litter input from NPP in expanded discussion and the mismatch between the trends in NPP (as proxy of the litter) and simulated SOC by adding Fig.S1. I have no further suggestions for the revision

**Author's response**

Thank you very much for the positive feedback on the revised manuscript!